# HARDWARE-AWARE COMPRESSION WITH RANDOM OPERATION ACCESS SPECIFIC TILE (ROAST) HASHING

## ABSTRACT

Advancements in deep learning are often associated with increasing model sizes. Training and deploying large models require sophisticated hardware and incur significantly higher costs. Thus, model compression is a widely explored approach to solving the problem. However, SOTA techniques fall short in one or more desirable aspects of compression - for instance, pruning does not reduce memory for training, quantization can only provide up to 32x compression, HashedNet is cache-inefficient, etc. This paper proposes a model-agnostic, cache-friendly, and hardware-aware model compression approach: Random Operation Access Specific Tile (ROAST) hashing. ROAST collapses the parameters by clubbing them through a lightweight mapping. While clubbing these parameters, ROAST utilizes cache hierarchies by aligning the memory access pattern with the parameter access pattern. ROAST is up to $\sim 25\times$ faster to train and $\sim 50\times$ faster to infer than the popular parameter sharing method HashedNet. Additionally, ROAST introduces global weight sharing, which is empirically and theoretically superior to local weight sharing in HashedNet, and can be of independent interest. With ROAST, we can efficiently train and deploy the model using a much smaller memory footprint ($\sim 10 - 100\times$ lesser) in text and image classification tasks.

## 1 INTRODUCTION

Models across different domains, including Natural Language Processing (NLP), Computer Vision (CV), and Information Retrieval (IR), are exploding in size. State-of-the-art (SOTA) results in these domains are being obtained at a disproportionate increase in model sizes, questioning the sustainability of deep learning (Thompson et al., 2021). For instance, SOTA architectures for vision include VGG (Simonyan & Zisserman, 2014) (150M params, 0.6GB) and ViT (Dosovitskiy et al., 2020) (up to 304M params, 1.2GB). Additionally, SOTA NLP models range from BERT (Devlin et al., 2018) (340M params, 1.36GB) to GShard (Lepikhin et al., 2020) (600B params, 2.4TB). Similarly, industrial-scale recommendation models such as DLRM (Naumov et al., 2019; Mudigere et al., 2021) can have up to 10s of trillions of parameters (50TB).

Large models, such as the above, come with various challenges. They need high-end distributed hardware for training and deployment, incurring higher costs. Additionally, the required model-parallel setup has higher inference and training-iteration latency for these models. Model compression is a research direction that aims to resolve these issues by reducing the memory footprint of the model. Compression of the order of $100\times$ can eliminate the need for model-parallel setup for many SOTA models like GPT(Radford et al., 2019), Gshard(Lepikhin et al., 2020), DLRM (Naumov et al., 2019) which now can fit on a single GPU. Furthermore, compressing large models to small sizes come with immediate latency benefits. For example, Desai et al. (2022) showed that by compressing the DLRM model $1000\times$ and using 1 GPU instead of 8 GPUs, we could get $3\times$ faster inference at a lower cost. Also, in the case of CPU inference, a smaller model is efficient. For example, (Diamos et al., 2016) showed that if a single RNN layer can fit in registers, it leads to $146\times$ faster inference.

Thus, the ML community has heavily invested in model compression. A variety of model compression paradigms now exist in literature like pruning (Han et al., 2016b), quantisation (Han et al., 2016b), knowledge distillation (Buciluǎ et al., 2006), parameter-sharing (Chen et al., 2015; Desai et al., 2022), and low rank decomposition (Hrinchuk et al., 2020; Yin et al., 2021). Table 1 compares these approaches on three considerations (1) if the model memory is reduced for training. (2) if the memory size can be controlled independently of the model, and (3) if the approach considers the underlying

Table 1: Various compression techniques on three aspects (1) Memory reduction during training ( apart from inference) (2) arbitrary control over memory (3) Hardware awareness / cache-efficiency * Some versions of pruning that are tuned to the underlying hardware and are cache-efficient

| | Training memory reduction | Arbitrary control on memory | Cache efficient |
|---|---|---|---|
| **Pruning** | No | No | No* |
| Low-rank decomposition | Yes | No | Yes |
| **Low-precision** | Yes | No | Yes |
| **Quantization aware training (QAT)** | No | No | N.A |
| **Parameter sharing - HashedNet** | Yes | Yes | No |
| **Knowledge Distillation** | No | No | N.A |
| **ROAST (ours)** | Yes | Yes | Yes |

memory hierarchies and is cache-efficient. We want the techniques to fare positively in these three aspects. However, techniques like pruning, QAT, and knowledge distillation require us to use the memory of the original model while training and only reduce inference time memory. Additionally, there are limits to compression obtained by quantization and pruning depending on which component we are compressing. For example, we cannot prune an embedding table ($N \times d$) more than $d\times$ as we do not want any embedding vector to have all zeros. HashedNet provides memory reduction during training and arbitrary control over memory. However, the look-ups in HashedNet are randomly and independently distributed across the total memory. This makes HashedNet cache-inefficient.

This paper presents Random Operation Access Specific Tile (ROAST) hashing, a parameter-sharing approach that provides cache-efficiency and arbitrary control over memory during training as well as inference. ROAST does not change the model's functional form and can be applied to all computational modules of a model, such as MLP layers, attention blocks, convolution layers, and embedding tables. ROAST is hardware aware: it proposes a tile-based hashing scheme tuned to the memory access pattern of the algorithmic implementation of the operation being performed. ROAST uses this hash function to recover blocks of the model from a single array of parameters - ROAST array. ROAST is superior to HashedNet due to two factors (1) Unlike HashedNet, ROAST proposes global weight-sharing where parameters are shared across the different computational modules. As we shall see, global weight-sharing is empirically and theoretically superior to local weight-sharing and might be of independent interest. (2) ROAST uses block-based hashing, which is theoretically superior to count-sketch hashing used in HashedNet. (Desai et al., 2022)

We show that with ROAST, we can train a BERT-2-2 ( 2 layers, 2 attention heads) model on the largest available text-classification datasets (amazon-polarity, yelp-polarity) using $100\times$ lesser memory without loss of quality. In cases where the model is overly parameterized, like using BERT-12-12 in the text classification task above, we can still obtain similar compression of $100\times$. Thus it is a good alternative to neural architecture search. The results extend to CV datasets as well. Specifically, we can train a ResNet-9 model with $10\times$ lesser memory for the CIFAR10 dataset. Importantly, we show that ROAST, due to its hardware-aware nature, is significantly faster than HashedNet: ROAST is up to $\sim 25\times$ faster to train and $\sim 50\times$ faster to infer than HashedNet for large matrix multiplications. Our current implementation of ROAST matrix multiplication is about $1.34\times$ slower than full matrix multiplication in pytorch. This is a testament to how optimized CUBLAS libraries are. We believe, with enough investigation, we can make ROAST-MM comparably efficient to pytorch-MM as well.

**Limitations of ROAST:** One of the goals of model compression, apart from reducing memory usage, is to reduce computational workload for deployment. ROAST, currently, is not devised to decrease computation; it only decreases the memory footprint of a model. Reducing computation with a small memory is left for future work. However, it should be noted that reducing the memory footprint can significantly reduce computation latency and power consumption. As shown in (Han et al., 2016a), accessing memory from RAM is $6400\times$ costlier than 32bit INT ADD and $128\times$ more expensive than on-chip SRAM access in terms of energy consumption. Additionally, RAM access generally is $\sim100\times$ slower than a floating-point operation. So, this model compression with ROAST, in our opinion, is an important step for efficient training and inference.

## 2 RELATED WORK

This section briefly reviews the rich history of model compression paradigms. Model compression can be generally classified into two categories: (1) Compressing a learned model and (2) Learning a compressed model. ROAST lies in the second category.

**Compressing learned models: 1) Pruning:** Pruning (Zhu & Gupta, 2017) is a technique to remove parts of a large model, including weights, blocks, and layers, to make the model lighter. Pruning can be performed as a one-time operation or gradually interspersed with training. **2) Quantization:** Quantization can involve reducing the precision of the parameters of a model. Mixed precision models are sometimes used where different precision is used with different weights. KMeans quantization is another type of quantization, where models' weights are clustered using KMeans, and each cluster's centroid is used for all cluster weights. Model compression, in this case, is achieved by reducing the number of distinct weights. **3) Knowledge distillation:** Knowledge distillation (Buciluǎ et al., 2006) is widely applied in model compression with a focus on distilled architectures. Knowledge distillation involves training a teacher model (large original model); then, a student model is trained using the logits of the teacher model. Empirically, the student model trained under this paradigm generalizes better than the student model trained standalone. Many variations exist on this basic idea of knowledge distillation.

While these techniques have successfully reduced memory for inference, one of the drawbacks of this line of compression is that the memory usage while training the model is not reduced. ROAST, however, provides a solution that reduces the model's memory during training and inference.

**Learning compressed models 1) Low-rank decomposition:** In this method, matrices in the model are decomposed into a product of two low-rank matrices, thus saving memory per matrix. A generalization of low-rank decomposition to tensors is called tensor-train decomposition **2) Parameter sharing:** Parameter sharing approaches such as HashedNet (Chen et al., 2015) are generally used for matrix compression. These approaches randomly share weights among different parameters, reducing the model's memory usage.

This line of research provides model reduction even during training. However, Low-rank decomposition does not offer arbitrary control over memory footprint, and HashedNets are inefficient due to heavy cache-trashing caused by non-local lookups. Conversely, ROAST is a model-agnostic parameter-sharing approach that can arbitrarily reduce the model size without affecting the functional form while keeping the model recovery efficient.

## 3 BACKGROUND

**HashedNet: Compressing MLP matrices** Previous work (Chen et al., 2015) introduced a weight sharing method to compress weight matrices of MLP models. They map each matrix parameter to a shared parameter array using a random hash function xxhash (Collet, 2016). In the forward pass, this mapping is used to recover a weight matrix and perform matrix multiplication for each MLP layer. In the backward pass, the gradients of each weight matrix are mapped to the shared compressed array and aggregated using the sum operation. It should also be noted that each MLP layer uses an independent array of parameters. One of the main concerns with HashedNet is that memory accesses on the compressed array are non-coalesced. Thus, fetching a compressed matrix via HashedNet requires significantly more memory read transactions than fetching an uncompressed matrix for which memory accesses can coalesce. Our evaluation shows that uncoalesced memory accesses lead to high latency, especially for large matrices.

**Random Block Offset Embedding Array (ROBE) for embedding compression** In ROBE (Desai et al., 2022), the embedding table is generated using an array of parameters. The embedding of a token is obtained by drawing chunks of the embedding from the ROBE array. The locations of the chunks are decided randomly via light-weight universal hash functions. Authors of ROBE showed that ROBE hashing is theoretically superior to feature hashing used in HashedNet. Also, the use of chunks causes memory accesses to coalesce, making embedding lookup efficient.

ROAST proposes a component agnostic, global parameter sharing approach that tunes the hashing function to match memory accesses of algorithmic implementation of operation over available hardware, thus giving a superior parameter sharing scheme.

## 4 RANDOM OPERATION ACCESS SPECIFIC TILE (ROAST) HASHING

Let $\mathcal{M}$ be the compressed memory from which parameters will be used, $f$ be the model or the function that we want to run using $\mathcal{M}$, and $W$ be the recovered weights used in $f$. $f$ can be considered as a composition of operations $\{\mathcal{O}_i(X_i, W_i)\}$. By operation, we mean the smaller functions that, when composed together, give us the model $f$. Here $X_i$ is the input to the operation, and $W_i$ is the weights (i.e., learnable parameters) that $\mathcal{O}_i$ uses. Generally, $W_i$s are distinct and do not share parameters.

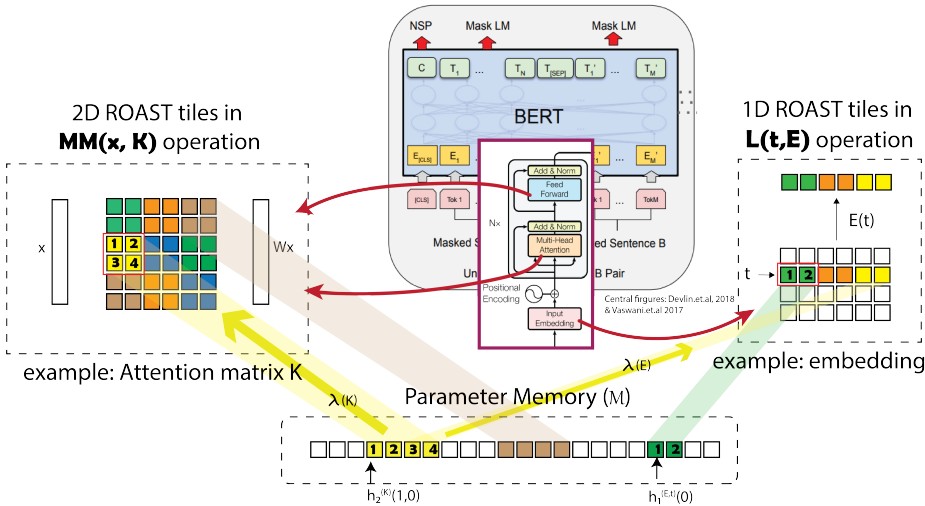

Figure 1: Generic model compression with operation-specific blocking for BERT as an example : (left) Shows how 2D tiles are mapped to $\mathcal{M}$ in case of **MM** operation. (right) Shows how 1D tiles are mapped to $\mathcal{M}$ in case of **L** operation. $\lambda$ is the module-specific GMS scaling factor

Random Operation Access Specific Tile (ROAST) hashing is a way to perform efficient model-agnostic parameter sharing-based compression. The following distinct aspects of ROAST set it apart from previous parameter sharing-based methods. (1) ROAST is a generic technique applicable to all computational modules. (2) ROAST proposes to tune its mapping from $W_i$ to $\mathcal{M}$ in a way that coalesces memory accesses according to how memory is accessed during the operation. This makes ROAST efficient and up to $45\times$ faster than competing approaches like HashedNet. (3) ROAST proposes Global Memory Sharing (GMS) as opposed to Local Memory Sharing (LMS) used in HashedNet. We show GMS to be theoretically and empirically superior to LMS in Section 5 and 6.

## 4.1 ROAST OPERATIONS IN DEEP LEARNING

Any model $f$ can be considered as a composition of smaller functions $\{\mathcal{O}_i(X_i, W_i)\}$. There are multiple ways to perform this decomposition depending upon what we consider a valid (or small enough) operation. In ROAST, we consider three types of operations: (1) $\mathbf{L}(l, W)$, lookup that accesses $\mathcal{M}$ and recovers $l^{th}$ element of $W$, say $w$. By element, we mean some particular part of $W$ that is identifiable by an integer. An example with embedding tables is given in figure 1. (2) $\mathbf{MM}(X, W)$, matrix multiplication that multiplies $X$ with $W$ and returns the result, and (3) $\mathbf{N}(X)$, various operations that only act on the input but do not interact with $\mathcal{M}$. In ROAST, in order to limit the memory usage, we make sure that $\mathbf{L}$ is used only on a small $w$ and $\mathbf{MM}$ is performed without recovering the entire matrix. We find that most deep learning models, if not all, can be written as a composition of operations $\mathbf{N}$, $\mathbf{MM}$ and $\mathbf{L}$, where $\mathbf{L}$ is only applied on small parameters. Let us discuss how ROAST implements $\mathbf{L}$ and $\mathbf{MM}$ operations in the following paragraphs.

**Lookup ($\mathbf{L}(l, W)$)** We recover a parameter weight $w$ of any shape in a row-major format. Thus, we can consider $w = W(l)$ to be a 1D vector without loss of generality. ROAST recovers $w$ from $\mathcal{M}$ in a blocked fashion. Consider $w$ to be composed of chunks of size $Z$. Each chunk $c$ is located in $\mathcal{M}$ using a universal hash function $h_1$ and is recovered from the location $h_1(c)$ in $\mathcal{M}$. Let $C(i)$ give the chunk number of index $i$ and $O(i)$ give the offset of $i$ in this chunk.

$$w[i] = \lambda \mathcal{M}[h_1(C(i)) + O(i)] \qquad h_1 : \mathbb{N} \to \{0, ..., |\mathcal{M}| - Z\} \qquad (1)$$

The recovered $W$ has $\lambda$ as a scaling factor discussed in section 4.2. The hash function hashes to a range $\{0, ..., |\mathcal{M}| - Z\}$ to avoid overflows while reading the memory. For example, Figure 1 (right) illustrates the embedding lookup using $\mathbf{L}$ with chunk size of 2. ROAST uses $\mathbf{L}$ to implement computational modules such as embeddings, bias vectors, and so on. We generalize the embedding lookup kernel from ROBE (Desai et al., 2022) to implement our $\mathbf{L}$ kernel.

**Matrix multiplication ($\mathbf{MM}(X_i, W_i)$)** 2D matrix multiplication is one of the most widely used operations in deep learning. We implement our ROAST-MM kernel with parameter sharing performed in a way that the algorithm for matrix multiplication accesses coalesced pieces of $\mathcal{M}$. An efficient

implementation of matrix multiplication on GPU follows a block multiplication algorithm to use the on-chip shared memory efficiently. While computing $C = A \times B$, A, B and C are divided in tiles of size $Z_0 \times Z_1$, $Z_1 \times Z_2$ and $Z_0 \times Z_2$ respectively. Thus, we divide our 2D weight matrix into tiles of size $Z_1 \times Z_2$. The tile, $(x, y)$, where $x$ and $y$ are the coordinates of the tile, is located in $\mathcal{M}$ in a row-major format via a universal hash function $h_2(x, y)$. Let $C_1(i, j)$ and $C_2(i, j)$ give the $x$-coordinate and $y$-coordinate of the tile to which $i, j$ belongs. Similarly, let $O_1(i, j)$ and $O_2(i, j)$ give the $x$-offset and $y$-offset of a location $(i, j)$ on the tile. Then, we use the following mapping for ROAST-MM,

$$W[i, j] = \lambda \mathcal{M}[h_2(C_1(i, j), C_2(i, j)) + Z_2 O_1(i, j) + O_2(i, j)]$$

$$h_2 : \mathbb{N}^2 \to \{0, ..., |\mathcal{M}| - Z_1 Z_2\}$$

Again, $\lambda$ is the scaling factor discussed in section 4.2. The hash function hashes to a range $\{0, ..., |\mathcal{M}| - Z_1 Z_2\}$ to avoid overflows while reading the chunk. Figure 1 (left) illustrates ROAST-MM with a chunk size of $2 \times 2$. The above mapping is used whenever a 2D tile is accessed in the matrix multiplication algorithm. The pseudo code for ROAST-MM is shown in algorithm 1. We talk about implementation of this kernel and its evaluation in the later part of the paper. ROAST uses ROAST-MM kernel to implement computational modules such as MLP layers, attention blocks, etc. Each module invoking ROAST kernels uses independent hash functions.

---

**Algorithm 1** ROAST-MM($I \times H \times O$)

---

**Require:** $X \in R^{I \times H}$, $\mathcal{M}$, $\lambda$, $h : \mathbb{N}^2 \to \{0, ..., |\mathcal{M}| - Z_1 Z_2\}$
**Ensure:** $output = \mathbf{MM}(X, \mathcal{M}[h(:, :)])$
   $value \leftarrow \mathbf{TILE}(Z_0, Z_2)$             ▷ Allocate a 2D tile of size $Z_0 \times Z_2$ to accumulate results
   **for** $i \in \{0, 1, ..., \lceil I/Z_0 \rceil - 1\}$ **do**
      **for** $j \in \{0, 1, ..., \lceil O/Z_2 \rceil - 1\}$ **do**
         $value[:, :] \leftarrow 0$
         **for** $k \in \{0, 1, ..., \lceil H/Z_1 \rceil - 1\}$ **do**
            $value \leftarrow value + \mathbf{MM}(X[i : i + Z_0, k : k + Z_1], \mathcal{M}(h(k : k + Z_1, j : j + Z_2)))$
                             ▷ Access to the weight tile passes through the hash function
         **end for**
         $output[i : i + Z_0, j : j + Z_2] \leftarrow \lambda * value$
      **end for**
   **end for**

---

Apart from scaling each recovered parameter with module-specifc $\lambda$, we can also multiply it with another independent hash function $g : \mathbb{N}^k \to \{\pm 1\}$ ($k$=1 or $k$=2).

## 4.2 GLOBAL MEMORY SHARING (GMS)

HashedNet uses local memory sharing (LMS), which states that each layer will have independent compressed memory. In contrast, ROAST proposes global memory sharing (GMS), wherein we share memory across modules. However, modules cannot directly use the parameters stored in $\mathcal{M}$ as each module's weights requires initialization and optimization at different scales. For instance, in the Xavier's initialization (Glorot & Bengio, 2010), weights are initialized with distribution $\mathbf{Uniform}(-1/\sqrt{n}, 1/\sqrt{n})$ where $n$ is size of the input to the module. In GMS, we must ensure that each module gets weights at the required scale. To achieve this, we first initialize the entire ROAST parameter array with values from the distribution $\mathbf{Uniform}(-1/C, 1/C)$ for some constant $C$. Then, for each module, we scale the weights retrieved from the ROAST array by a factor of $\lambda = C/\sqrt{n}$.

One can understand the benefit of GMS over LMS in terms of the number of distinct functions in $f$ that can be expressed using a fixed $\mathcal{M}$. Consider a family of functions with $n$ parameters. GMS can potentially express $|\mathcal{M}|^n$ functions across different random mappings. In LMS, let separate parameters be of sizes $n_1, n_2, ..n_k$ and each of them is mapped into memories $\mathcal{M}_1, \mathcal{M}_2, ..., \mathcal{M}_k$. Thus, $n = \sum_i n_i$ and $|\mathcal{M}| = \sum_i |\mathcal{M}_i|$. Then LMS can only express $|\mathcal{M}_1|^{n_1} |\mathcal{M}_2|^{n_2} .... |\mathcal{M}_k|^{n_k}$ different functions. Thus expressivity of LMS is strictly less than that of GMS and can be orders of magnitude less depending on exact values of $n_i$ and $|\mathcal{M}_i|$. We also show that GMS is superior to LMS in terms of dimensionality reduction (feature hashing) in Section 5.

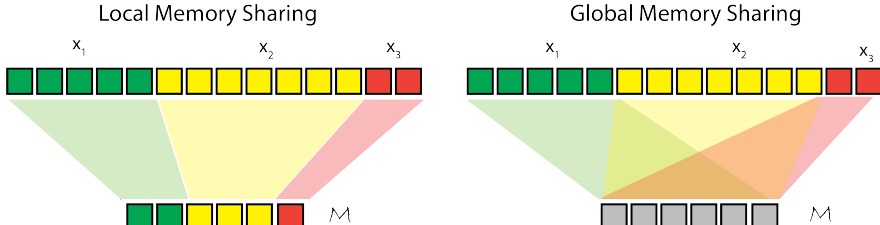

Figure 2: Local memory sharing : each module compresses its parameters separately. In Global memory sharing, all parameters from accross the modules share the same memory

### 4.3 FORWARD AND BACKWARD PASSES

Recall that in ROAST, operations are of three types $\mathbf{L}, \mathbf{MM}$ and $\mathbf{N}$. The forward pass proceeds by applying each operation in sequence. If an operation is of type $\mathbf{N}$, we directly apply its function on the input. For $\mathbf{L}$ and $\mathbf{MM}$ operations, outputs are computed according to the procedure described in Section 4.1.

The gradient of the loss w.r.t a weight $w_i$ in $\mathcal{M}$ is the $\lambda$-scaled aggregation of gradients of loss w.r.t all the parameters that map to this weight. For simplicity of notation, consider $\theta$ as the complete parameter, $\lambda(j)$ as the scaling factor we use for the module that $\theta_j$ belongs to, and $h$ be the mapping from $\theta$ to $\mathcal{M}$. See Appendix B.1 for more details.

$$\nabla_{w_i} f(w) = \sum_{j, h(j)=i} \lambda(j) * \nabla_{\theta_j} f(\theta) \tag{2}$$

### 4.4 IMPLEMENTATION OF ROAST-MM

The high-performance community has heavily investigated the fast implementation of the General Matrix Multiplication (GEMM) kernel, a fundamental operation in many computational workloads, including deep learning. Optimized implementations of GEMM kernels are available in vendor libraries such as cuBLAS (NVIDIA Corporation, 2022a) and CUTLASS (NVIDIA Corporation, 2022b). Unfortunately, these implementations do not support custom tile loading operations, which is the key of ROAST-MM. To implement ROAST-MM to a reasonable level of efficiency, we used Triton (Tillet et al., 2019): an intermediate language for tiled neural network computations. Triton abstracts out the shared memory management to make it helpful in customizing tiled operations with high efficiency.

In our implementation of ROAST-MM, the optimal size of coalesced tiles is a parameter that depends on the shape of the weight matrix. Therefore, different tile sizes can lead to different parallelism, occupancy, and shared memory efficiency, resulting in different execution times. We autotune this parameter to obtain the best performance for particular matrix shapes. We propose two strategies for autotuning each ROAST-MM layer - (1) Optimize the inference workload by autotuning the forward kernel and sharing the tile size with the backward kernels. (2) Optimize the training workload by autotuning the forward and backward kernels together. Extensive evaluation of this kernel is presented in appendix C.2.

## 5 FEATURE HASHING QUALITY: GLOBAL MEMORY SHARING ADVANTAGE OVER LOCAL MEMORY SHARING

We can consider model compression as dimensionality reduction of a parameter vector (a one dimensional vector of all parameters in a model) of size $n$ into a vector of size $|\mathcal{M}| = m$. Quality of inner-product preservation is used as a metric to measure the quality of dimensionality reduction. In terms of dimensionality reduction, ROAST uses ROBE hashing, which shows that chunk based hashing is theoretically better than hashing individual elements. In this section, we compare ROAST's GMS proposal against HashedNet's LMS using a chunck size of one. Consider two parameter vectors $x, y \in R^n$, we are interested in how the inner product of parameter vectors are preserved under hashing. Let $x = [x_1, x_2, ..., x_k]$ and $y = [y_1, y_2, ..., y_k]$ be composed of $k$ vectors of sizes $n_1, n_2, ...n_k$ where [] denotes concatenation. In LMS, let each piece map to memory of size $f_i m$ where $\sum_i f_i = 1$. The estimated inner product with GMS is

$$\widehat{\langle x, y \rangle}_{G,m} = \sum_{j=1}^{m} \left( \sum_{i=1}^{n} \mathbb{I}(h(i)=j)g(i)x[i] \sum_{i=1}^{n} \mathbb{I}(h(i)=j)g(i)y[i] \right) \tag{3}$$

Table 2: Experimental settings: The datasets and models used in experiments.

| Domain | Task | Dataset | #Samples | Model | Model size |
|---|---|---|---|---|---|
| NLP | text-classification | amazon-polarity | 3.6M/0.4M | BERT-2-2 | 37.43M |
| NLP | text-classification | yelp-polarity | 560K/38K | BERT-2-2 | 37.43M |
| CV | image-classification | cifar10 | 50K/10K | ResNet | 6.5M |

The estimated inner product with LMS can be written as

$$\widehat{\langle x, y\rangle}_{L,m,\vec{f}} = \sum_{l=1}^{k}\sum_{j=1}^{f_l m}\left(\sum_{i=1}^{n_l}\mathbb{I}(h(i){=}j)g(i)x_l[i]\sum_{j=1}^{n_l}\mathbb{I}(h(i){=}j)g(i)y_l[i]\right) = \sum_{l=1}^{k}\widehat{\langle x_l, y_l\rangle}_{G,(f_l m)}$$

(4)

**Theorem 1** *Let $x, y \in R^n$ and be composed of $k$ vectors $x = [x_1, x_2, ..., x_k]$ and $y = [y_1, y_2, ..., y_k]$. Then the inner product estimation of global and local weight sharing are unbiased.*

$$\mathbb{E}(\widehat{\langle x, y\rangle}_{G,m}) = \langle x, y\rangle \qquad \mathbb{E}(\widehat{\langle x, y\rangle}_{L,m,\vec{f}}) = \langle x, y\rangle$$

(5)

*The variance for inner product estimation can be written as,*

$$\mathbb{V}_G(\widehat{\langle x, y\rangle}) = \sum_i f_i V_i + \frac{1}{m}\left(\sum_{i,j,i\neq j}(||x_i||^2||y_j||^2) + \langle x_i, y_i\rangle\langle x_j, y_j\rangle\right)$$

(6)

$$\mathbb{V}_L(\widehat{\langle x, y\rangle}) = \sum_i V_i$$

(7)

*where*

$$V_l = \frac{1}{f_l}\frac{1}{m}\left(\sum_{i\neq j}a_i^2 b_j^2 + \sum_{i\neq j}a_i b_i a_j b_j\right), \text{ where } x_l = (a_1, a_2..., a_{n_l}) \text{ and } y_l = (b_1, b_2..., b_{n_l})$$

(8)

*where $\mathbb{V}_L$ is local memory sharing variance and $\mathbb{V}_G$ is global memory sharing variance.*

**Intuition:** The two terms in $\mathbb{V}_G$ can be understood as follows: The first term is the local variance with individual terms reduced by a factor of $f_i$. This is because each piece of the vector is being distributed in a memory that is $1/f_i\times$ larger. However, in GMS, there is a possibility of more collisions across pieces. This leads to the second term in $\mathbb{V}_G$. Note that, for a given $x, y$ and a finite value for $m$, $\mathbb{V}_G$ is always bounded. At the same time, $\mathbb{V}_L$ is unbounded due to $0 < f_i < 1$ in the denominator. So if the number of pieces increases or particular $f_i$ grows smaller, $\mathbb{V}_L$ increases. While we cannot prove that $\mathbb{V}_G$ is strictly less than $\mathbb{V}_L$, we can investigate the equation under some assumptions on the data. Practically, each piece of the parameter vector is a computational block like a matrix for multiplication or embedding table lookup. These blocks are initialized at a scale proportional to the square root of their size. So the norms of these vectors are similar. Let us assume the norm of each piece to be $\sqrt{\alpha}$. Also, let us assume that over random data distributions over $x$ and $y$, all the inner products to be $\beta$ in expectation. Then,

$$\mathbb{V}_G \approx \frac{k^2}{m}(\alpha^2 + \beta^2) \qquad \mathbb{V}_L \approx \frac{1}{m}(\alpha^2 + \beta^2)(\frac{1}{f_1} + \frac{1}{f_2} + ... + \frac{1}{f_k}) \geq \frac{1}{m}(\alpha^2 + \beta^2)k^2\frac{1}{(\sum f_i)} = \mathbb{V}_G$$

(9)

Thus, $V_L$ is greater than $V_G$, and it can be much greater depending on the exact values of $f_i$. The proof of the theorem and other details are presented in Appendix B.2

## 6 EXPERIMENTAL EVALUATION

**Setup:** In this section, we evaluate the ROAST compression approach on two types of tasks. The details of the tasks, datasets and models used are mentioned in table 2. . For image-classification tasks, we choose the cifar-10 dataset and the leader for the DawnBenchmark (Coleman et al., 2017) - a ResNet-9 model[1] for cifar-10. The target accuracy for this benchmark is 94% and hence we perform hyper-parameter tuning to get a test accuracy of $\geq 94\%$. We stop the tuning once we

---

[1]https://github.com/apple/ml-cifar-10-faster

Table 3: Text classification task. (above) ROAST shows up to $100\times$ compression without loss of quality on BERT-2-2 model. (below) Even in case of overparameterized model of BERT-12-12, ROAST is able to maintain the quality similar compression.

| Text-classification Acc | | | |
|---|---|---|---|
| **Model** | **size** | **amazon-polarity** | **yelp-polarity** |
| BERT-2-2 | 37.4M | 93.4 | 90.8 |
| BERT-1-1 | 30.3M | 92.01 | 90.2 |
| ROAST-10x (BERT-2-2) | 3.7M | 94.6 | 91.1 |
| ROAST-100x (BERT-2-2) | 393K | 93.8 | 91.0 |
| BERT-12-12 (BERT-base) | 108M | 93.5 | 90.8 |
| ROAST-10x (BERT-base) | 10.1M | 94.64 | 90.9 |
| ROAST-100x (BERT-base) | 1.1M | 93.9 | 90.8 |

Table 4: Image classification task: (above)We see that ResNet-9 model can be trained in $10\times$ smaller memory. (below). Pruning gives $100\times$ post-training compression but requires complete memory for training. We can prune ROAST-10x model, which uses $10\times$ lesser memory, further $10\times$ to give $100\times$ post-training model

| Image-classification Acc (target: 94%) | | |
|---|---|---|
| **Model** | **Size** | **cifar-10** |
| ResNet-9 | 6.5M | 94.2 |
| ROAST-5x | 1.2M | 94.58 |
| ROAST-10x | 650K | 94.15 |
| PRUNE-10x | 650K (6.5M) | 95.59 |
| PRUNE-100x | 65K (6.5M) | 94.8 |
| PRUNE-1000x | 6.5K (6.5M) | 93.34 |
| ROAST-10x- PRUNE-10x | 65K (650K) | 94.06 |

reach this accuracy and hence the results for CIFAR-10 should be compared w.r.t whether it crosses 94.0%. For NLP tasks, we use two largest available text-classification datasets on huggingface (HuggingFace, 2022). For the model, we use BERT-x-y (x:number of layers, y:number of attention heads) architecture with classification head. On both NLP datasets, using models larger than BERT-2-2 lead to similar test accuracy and hence we choose BERT-2-2 as the base model. The other hyper parameters for NLP tasks are { batch 64 for amazon-polarity and 32 for yelp-polarity, learning rate 2e-5, AdamW optimizer, Linear scheduler}

**Roast for compression** As we can see in tables 3 and 4 , with ROAST, we can achieve similar quality of model in much smaller space. Specifically, for text-classification, we see that we can train and deploy the BERT-2-2 model in $100\times$ lesser space. Similarly, we can train and deploy ResNet model in $10\times$ lesser space for same target test accuracy. Thus, ROAST is an effective method for training and deploying models on memory-constrained systems.

**Managing excess parameters** It is clear from table 3, that BERT-base architecture is highly over parameterized for the tasks under consideration. However, even in this case, ROAST can be used to control the memory footprint while maintaining the functional form of the larger model.

**Pruning and ROAST** We perform unstructured iterative-magnitude pruning (Han et al., 2016b) on ResNet model and find that pruning gives upto $100\times$ compression. However note that pruning requires us to train the model using memory required to store the original model. However, compression with ROAST means using lesser memory even for training. Additionally, pruning can be used in conjunction with ROAST to obtain smaller models using smaller memory. In table 4, we see that we can prune 90% of weights in $10\times$ compressed ROAST array and still achieve the same quality.

**Local vs. Global memory sharing** In the figure 3, we show that the quality of the model while using global memory sharing is, indeed, better than local memory sharing. This supports our theoretical observation about these memory sharing schemes.

**Efficiency of ROAST operators as compared to HashedNet** Table 7 shows the inference performance of a simple model using ROAST-MM for matrix multiplication on compressed memory. Our model linearly transforms the input vector and computes its norm. We optimized the ROAST-MM kernel for this experiment using the inference-optimal strategy. We make the following observations

Table 5: Inference times of different square weight matrices using an input batch of 512. For ROAST, the tile parameters of each matrix multiplication are autotuned. The measurements were taken using TF32 on a NVIDIA A100 GPU (48GB). We used PyTorch's matmul function (MM) for the full uncompressed matrix multiplication. ■:bad ■: good

| Inference time (ms) | | | | | | | | | |
|---|---|---|---|---|---|---|---|---|---|
| | | Weight matrix dimensions (Dim × Dim) | | | | | | | |
| Model | $\mathcal{M}$ size ↓ | 512 | 1024 | 2048 | 4096 | 8096 | 10240 | 20480 | Average |
| Full size → | | 1MB | 4MB | 16MB | 64MB | 128MB | 420MB | 1.6GB | |
| PyTorch-MM | | 0.10 | 0.11 | 0.12 | 0.22 | 0.69 | 1.18 | 3.91 | 0.91 |
| HashedNet | 4MB | 0.31 | 0.34 | 0.63 | 2.02 | 6.20 | 9.67 | 35.22 | 7.77 |
| | 32MB | 0.31 | 0.41 | 0.86 | 3.64 | 13.66 | 22.11 | 92.40 | 19.06 |
| | 64MB | 0.31 | 0.46 | 1.09 | 6.47 | 31.21 | 42.45 | 178.07 | 37.15 |
| | 128MB | 0.31 | 0.60 | 1.62 | 9.10 | 34.62 | 56.03 | 229.31 | 47.37 |
| | 256MB | 0.32 | 0.62 | 1.82 | 10.25 | 38.28 | 62.67 | 256.22 | 52.88 |
| | 512MB | 0.33 | 0.68 | 2.05 | 10.59 | 40.55 | 65.74 | 272.23 | 56.03 |
| ROAST | 4MB | 0.28 | 0.30 | 0.27 | 0.48 | 0.99 | 1.36 | 4.83 | 1.22 |
| | 32MB | 0.28 | 0.29 | 0.27 | 0.44 | 1.01 | 1.38 | 4.88 | 1.22 |
| | 64MB | 0.28 | 0.29 | 0.27 | 0.44 | 1.00 | 1.40 | 4.93 | 1.23 |
| | 128MB | 0.30 | 0.27 | 0.27 | 0.45 | 1.01 | 1.39 | 4.91 | 1.23 |
| | 256MB | 0.30 | 0.27 | 0.27 | 0.44 | 1.01 | 1.40 | 4.90 | 1.23 |
| | 512MB | 0.30 | 0.30 | 0.27 | 0.45 | 1.02 | 1.39 | 4.95 | 1.24 |

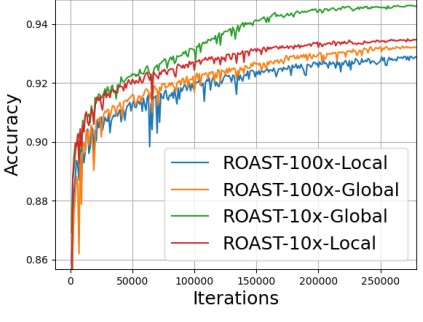

(a) GMS vs. LMS (amazon-polarity)

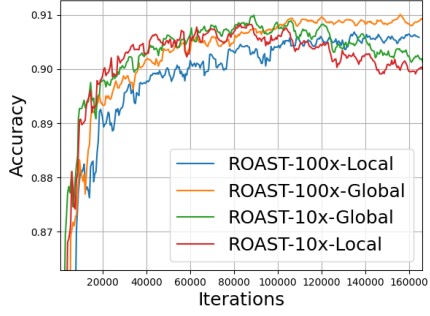

(b) GMS vs. LMS (yelp-polarity)

Figure 3: Effect of local and global memory sharing with compression of BERT-12-12 model for text-classification tasks. In yelp, rolling mean of 5 measurements is taken to reduce noise in plots

from Table 7: (1) ROAST-MM outperforms HashedNet kernel consistently across the different multiplication workloads. On an average over different workloads, ROAST-MM is up to $45\times$ faster than HashedNet. (2) ROAST-MM is $1.34\times$ slower than PyTorch-MM. This is expected as Pytorch-MM uses extremely optimized libraries for matrix multiplication and ROAST-MM implementation is comparatively under-optimized. It is still interesting to note that ROAST-MM's performance better in terms of scaling efficiency than PyTorch-MM with the increase in workload. As the workload increases $1600\times$ (from $512\times512$ to $20480\times20480$), PyTorch-MM takes $39\times$ time, HashedNet takes $106\times$ time whereas ROAST-MM only takes around $16\times$ time. We present more detailed measurements across different optimizers for training-optimal strategy in the appending C.2

## 7 CONCLUSION

Traditionally model compression has focused on memory reduction during inference. However, model memory during training is also an important consideration. While some of the existing methods such as HashedNet and Low-rank factorisation provide model reduction during training, these methods either do not provide cache-efficient model recovery or have implicit cap on memory reduction. ROAST overcomes these obstacles and provides a cache-efficient, arbitrary control over the memory footprint of model during training and inference. ROAST, essentially provides a practical parameter sharing method. ROAST is theoretically better than HashedNet in terms of dimensionality reduction due to block based hashing and global memory sharing. We empirically validate the efficiency advantage of ROAST over HashedNet and that we can achieve high compression with ROAST.

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

# A    ADDITIONAL DATA FOR REVIEWERS - PARTS OF WHICH WILL GO IN MAIN PAPER IN FINAL VERSION

## A.1    EXTENDED TABLE 3 WITH EPOCH INFORMATION AND MORE BASELINES

Table 6: Table 3 extended version

| Text-classification Acc | | | | | | |
|---|---|---|---|---|---|---|
| Model | size | amazon-polarity | Epochs to reach the acc | yelp-polarity | Epochs to reach the acc | Comment |
| BERT-2-2 | 37.4M | 93.4 | 5.6 | 90.8 | 5.4 | |
| BERT-1-1 | 30.3M | 92.01 | 7.02 | 90.2 | 2.8 | |
| ROAST-10x-GMS (BERT-2-2) | 3.7M | 94.6 | 7.3 | 90.8 | 2.8 | |
| ROAST-100x-GMS (BERT-2-2) | 393K | 93.8 | 7.2 | 90.8 | 7.03 | |
| PRUNE-10x (BERT-2-2) | 3.74M | 93.5 | 9.02 | 89.65 | 9 | full-9-1 schedule |
| PRUNE-100x(BERT-2-2) | 374K | 91.36 | 9.8 | 89 | 9.8 | full-9-1 schedule |
| PRUNE-10x(BERT-2-2) | 3.74M | 93.24 | 8.94 | 89.8 | 7 | full-1-9-schedule |
| PRUNE-100x(BERT-2-2) | 370K | 90.73 | 9.15 | 87.7 | 9.82 | full-1-9-schedule |
| BERT-12-12 | 108M | 93.51 | 6.95 | 90.8 | 4.7 | |
| BERT-12-12-10x-LMS | 10.1M | 93.49 | 4.84 | 90.9 | 4.69 | |
| BERT-12-12-10x-GMS | 10.1M | 94.64 | 4.85 | 91.1 | 4.97 | |
| BERT-12-12-100x-LMS | 10.1M | 92.9 | 4.87 | 90.7 | 9.03 | |
| BERT-12-12-100x-GMS | 10.1M | 93.9 | 9.39 | 91.0 | 6.83 | |
| Text-classification convergence for a specific target accuracy | | | | | | |
| Model | size | amazon-polarity | | yelp-polarity | | Comment |
| | | target | epochs | target | epochs | |
| BERT-12-12 | 108M | 93.4 | 5 | 90.8 | 4.7 | |
| BERT-12-12-10x-LMS | 10.1M | 93.4 | 3.77 | 90.9 | 4.69 | |
| BERT-12-12-10x-GMS | 10.1M | 93.4 | 1.97 | 90.9 | 3.4 | |
| BERT-12-12-100x-LMS | 10.1M | 92.9 | 4.78 | 90.7 | 9.09 | |
| BERT-12-12-100x-GMS | 10.1M | 92.9 | 3.09 | 90.7 | 3.74 | |

We add a lot of information and new results to the table. Specifically,

- We add the GMS and LMS results to the table separately. So that readers can get an idea of each of the method on the task.

- We add unstructured pruning (best pruning quality wise) resutls for NLP tasks as well. The pruning results are obtained in the following manner. With the full-9-1 schedule, we start from the fully trained model, perform iterative pruning during next 9 epochs and then tune the final pruned model for 1 more epoch. Whereas in the full-1-9 schedule, we again start from the fully trained model, perform pruning in the next 1 epoch and then tune the model further for 9 epochs. We note the best achieved accuracy with the final model structure and the epoch at which this accuracy is reached.

- For each result, we note the number of epoch when the best accuracy was reached.

- We append an additional small table which notes the number of epochs required to reach a target accuracy to compare the convergence of different models.

We make the following observations.

- GMS reaches better accuracy than LMS for the same amount of compression for both the datasets. Additionally, GMS reaches the same target accuracy faster than the LMS.

- The ROAST approach is more effective than pruning approaches in NLP tasks of text-classification for architectures like BERT.

- It is interesting that GMS-10× converges faster than original model on both datasets. We leave investigating this as future work.

## A.2    GMS VS LMS FOR YELP

As can be seen from the two plots in figure4, it is clear the GMS performs superior to LMS in both the compression settings.

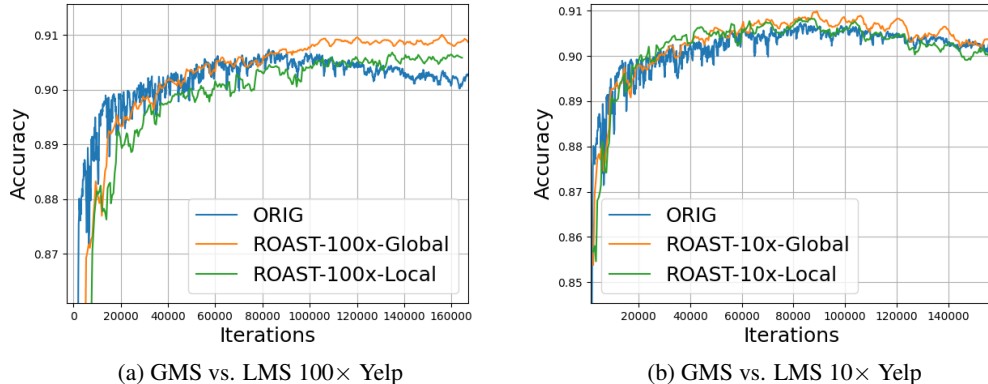

(a) GMS vs. LMS $100\times$ Yelp

(b) GMS vs. LMS $10\times$ Yelp

Figure 4: Separate plots for $10\times$ and $100\times$ ROAST for Yelp dataset for better visibility. Also, rolling mean of 5 measurements was used to reduce noise in the plots

# B  THEORY

ROAST is a generalized model compression which performs operation specific system-friendly lookup and global memory sharing. This raises some interesting theoretical questions

## B.1  BACKWARD PASS FOR MODEL SHARING WEIGHTS ACROSS DIFFERENT COMPONENTS

A general function sharing a weight, say $x$ across different components can be written as , $f(x, g(x))$ The interpretation is that x was used in g(.) and then again used ahead in f. (In case of MLP, we can think of x being used in multiple layers)

Let $f(g_1, g_2)$ where both $g_1$ and $g_2$ are functions of $x$.

$$\frac{\partial f(g_1, g_2)}{\partial x} = \frac{\partial f(g_1, g_2)}{\partial g_1} * \frac{\partial g_1}{\partial x} + \frac{\partial f(g_1, g_2)}{\partial g_2} * \frac{\partial g_2}{\partial x} \tag{10}$$

$g_1 = x$ and $g_2 = g(x)$

$$\frac{\partial f(g_1, g_2)}{\partial x} = \frac{\partial f(x, g(y))}{\partial x}\Big|_{y=x} + \frac{\partial f(y, g(x))}{\partial g(x)} * \frac{\partial g(x)}{\partial x}\Big|_{y=x} \tag{11}$$

$$\frac{\partial f(g_1, g_2)}{\partial x} = \frac{\partial f(x, g(y))}{\partial x}\Big|_{y=x} + \frac{\partial f(y, g(x))}{\partial x}\Big|_{y=x} \tag{12}$$

Renaming,

$$\frac{\partial f(x, g(x))}{\partial x} = \frac{\partial f(z, g(y))}{\partial z}\Big|_{y=x, z=x} + \frac{\partial f(z, g(y))}{\partial y}\Big|_{y=x, z=x} \tag{13}$$

Thus, we can essentially consider each place where x appears as new variables and then gradient w.r.t x is just summation of partial derivatives of the function w.r.t these new variables. Thus, it is easy to implement this in the backward pass. In order to make sure that the memory utilization in backward pass is not of the order of the recovered model size, we do not use the auto-differentiation of tensorflow/pytorch. We implement our own backward pass and it can be found in the code.

## B.2  GLOBAL FEATURE HASHING VS LOCAL FEATURE HASHING.

We can consider model compression techniques as dimensionality reduction of the parameter vector (a one dimensional vector of all parameters in a model) of size n into a vector of size $|\mathcal{M}| = m$. Quality of inner-product preservation is used as a metric to measure the quality of dimensionality reduction. In terms of dimensionality reduction, ROAST uses ROBE hashing Desai et al. (2022), which showed that chunk based hashing is theoretically better than hashing individual elements. In

this section, we analyse GMS proposal of ROAST against LMS of HashedNet. For the purpose of this comparison we assume a chunk size of 1. Consider two parameter vectors $x, y \in R^n$. We are interested in how inner product between these parameter vectors are preserved under hashing. Let $x = [x_1 x_2 ... x_k]$ and $y = [y_1 y_2 ... y_k]$ be composed of k pieces of sizes $n_1, n_2, ... n_k$. In LMS, let each piece be mapped into memory of size $f_i m$ where $\sum_i f_i = 1$.

The estimators of inner product in the GMS case can be written as ,

$$\widehat{\langle x, y \rangle}_{G,m} = \sum_{j=1}^{m} (\sum_{i=1}^{n} \mathbb{I}(h(i){=}j)g(i)x[i])(\sum_{i=1}^{n} \mathbb{I}(h(i){=}j)g(i)y[i]) \tag{14}$$

The estimate of inner product with LMS can be written as,

$$\widehat{\langle x, y \rangle}_{L,m,\vec{f}} = \sum_{l=1}^{k} \sum_{j=1}^{f_l m} (\sum_{i=1}^{n_l} \mathbb{I}(h(i){=}j)g(i)x_l[i])(\sum_{j=1}^{n_l} \mathbb{I}(h(i){=}j)g(i)y_l[i]) = \sum_{l=1}^{k} \widehat{\langle x_l, y_l \rangle}_{G,(f_i m)} \tag{15}$$

Note that

$$\widehat{\langle x, y \rangle}_{L,m,\vec{f}} = \sum_{l=1}^{k} \widehat{\langle x_l, y_l \rangle}_{G,(f_l m)} \tag{16}$$

The GMS estimator is the standard feature hashing estimator and the LMS is essentially sum of GMS estimators for each of the piece. as $E[g(i)] = 0$, it is easy to check by linearity of expectations that **Expectation** The suffix L refers to local hashing and G refers to global hashing.

$$E_G = \mathbb{E}(\widehat{\langle x, y \rangle}_{G,m}) = \langle x, y \rangle \tag{17}$$

$$E_L = \mathbb{E}(\widehat{\langle x, y \rangle}_{L,m,\vec{f}}) = \langle x, y \rangle \tag{18}$$

Let us now look at the variance. Let us follow the following notation,

- $V_G = \mathbb{V}(\widehat{\langle x, y \rangle}_{G,m})$. GMS variance of entire vectors
- $V_L = \mathbb{V}(\widehat{\langle x, y \rangle}_{L,m,\vec{f}})$. LMS variance of entire vectors
- $V_l = \mathbb{V}(\widehat{\langle x_l, y_l \rangle}_{G,f_l m})$. variance of each piece

we can write $V_l$ as follows. The following equation is easy to derive and it can be found the lemma 2 of Weinberger et al. (2009)

$$V_l = \frac{1}{f_l}\frac{1}{m}(\sum_{i \neq j} a_i^2 b_j^2 + \sum_{i \neq j} a_i b_i a_j b_j) \text{ where } x_l = (a_1, a_2 ... a_{n_l}) \text{ and } y_l = (b_1, b_2 ... b_{n_l}) \tag{19}$$

As, each of the piece is independently hashed in LSM, we can see

$$V_L = \sum_{l=1}^{k} V_l \tag{20}$$

Let us now look at $V_G$. Again, using lemma 2 from Weinberger et al. (2009)

$$V_G = \frac{1}{m}(\sum_{i \neq j} x_i^2 y_j^2 + \sum_{i \neq j} x_i y_i x_j y_j) \tag{21}$$

The expression can be split into terms that belong to same pieces and those across pieces

$$V_G = \frac{1}{m} \sum_{l=1}^{k} (\sum_{i \neq j \in \text{piece-l}} x_i^2 y_j^2 + \sum_{i \neq j \in \text{piece-l}} x_i y_i x_j y_j)$$

$$+ \frac{1}{m} \sum_{l1=1}^{k} \sum_{l2=1, l1 \neq l2}^{k} (\sum_{i \in \text{piece-l1}, j \in \text{pieces-l2}} (x_i^2 y_j^2) + \sum_{i \in \text{piece-l1}, j \in \text{pieces-l2}} x_i y_i x_j y_j))$$

$$V_G = \sum_{l=1}^{k} f_l V_l + \frac{1}{m} \sum_{l1=1}^{l} \sum_{l2=1, l1 \neq l2}^{l} ||x_{l1}||_2^2 ||y_{l2}||_2^2 + \langle x_{l1}, y_{l2} \rangle \langle x_{l2}, y_{l2} \rangle \tag{22}$$

**Observation 1:** In $V_L$ we can see that there are terms with $\frac{1}{f_l}$ which makes it unbounded. It makes sense as if number of pieces increase a lot a lot of compressions will not work for example if number of peices $> |\mathcal{M}|$. Also, it will affect $V_L$ a lot when some $f_l$ is very small which can often be the case. For example, generally embedding tables in DLRM model are much larger than that of matrix multiplciation modules (MLP) . which can make $f \approx 0.001$ for MLP components.

**Observation 2:** Practically we can assume each piece, no matter the size of the vector, to be of same norm. The reason lies in initialization. According to Xavier's initialization the weights of a particular node are initialized with norm 1. So for now lets assume a more practical case of all norms being equal to $\sqrt{\alpha}$. Also, in order to make the comparisons we need to consider some average case over the data. So let us assume that under independent randomized data assumption, the expected value of all inner products are $\beta$. With this , in expectation over randomized data, we have

$$V_G = \sum f_l V_l + \frac{k(k-1)}{m}(\alpha^2 + \beta^2) \tag{23}$$

Now note that,

$$V_l = \frac{1}{f_l}\frac{1}{m}\left(\sum_{i \neq j} a_i^2 b_j^2 + \sum_{i \neq j} a_i b_i a_j b_j\right) \text{ where } x_l = (a_1, a_2...a_{n_l}) \text{ and } y_l = (b_1, b_2...b_{n_l}) \tag{24}$$

(dropping the subscript "l" below)

$$V_l = \frac{1}{f_l}\frac{1}{m}\left((||x||_2^2 ||y||_2^2 + \langle x, y \rangle^2) - 2\sum_i x_i^2 y_i^2\right) \tag{25}$$

$$V_l = \frac{1}{f_l}\frac{1}{m}\left((\alpha^2 + \beta^2) - 2\sum_i x_i^2 y_i^2\right) \tag{26}$$

Note that for each negative term, there are $n_l$ positive terms. To simplify we disregard this term in the equation above. This is an approximation which is practical and only made to get a sense of $V_L$ and $V_G$ relation.

$$V_L - V_G = \sum V_l - \sum f_l V_l - \frac{k(k-1)}{m}(\alpha^2 + \beta^2)$$

$$V_L - V_G = \sum_l \frac{1}{m}(\frac{1}{f_l} - 1)((\alpha^2 + \beta^2)) - \frac{k(k-1)}{m}(\alpha^2 + \beta^2)$$

$$V_L - V_G = \sum_l \frac{1}{m}(\frac{1}{f_l} - 1)((\alpha^2 + \beta^2) - \frac{k(k-1)}{m}(\alpha^2 + \beta^2)$$

$$V_L - V_G \geq \frac{k(k-1)}{m}((\alpha^2 + \beta^2) - \frac{k(k-1)}{m}(\alpha^2 + \beta^2)$$

$$V_L - V_G \geq 0$$

Note that we ignored a term which reduces the $V_L$ a bit, Let the error be $\epsilon$

$$V_L - V_G \geq -\epsilon \tag{27}$$

The above equation shows even for the best case, $V_G$ might be slightly more than $V_L$. However for general case where harmonic mean is much worse than arithmetic mean, $V_L$ will be much larger depending on exact $f_l$ s

Table 7: Inference times of different square weight matrices using an input batch of 512. For ROAST, the tile parameters of each matrix multiplication are autotuned. The measurements were taken using TF32 on a NVIDIA A100 GPU (48GB). We used PyTorch's matmul function (MM) for the full uncompressed matrix multiplication. 🟥:bad 🟦: good

| Inference time (ms) | | | | | | | | | |
|---|---|---|---|---|---|---|---|---|---|
| | | Weight matrix dimensions (Dim × Dim) | | | | | | | |
| Model | $\mathcal{M}$ size ↓ | 512 | 1024 | 2048 | 4096 | 8096 | 10240 | 20480 | Average |
| Full size → | | 1MB | 4MB | 16MB | 64MB | 128MB | 420MB | 1.6GB | |
| PyTorch-MM | | 0.10 | 0.11 | 0.12 | 0.22 | 0.69 | 1.18 | 3.91 | 0.91 |
| HashedNet | 4MB | 0.31 | 0.34 | 0.63 | 2.02 | 6.20 | 9.67 | 35.22 | 7.77 |
| | 32MB | 0.31 | 0.41 | 0.86 | 3.64 | 13.66 | 22.11 | 92.40 | 19.06 |
| | 64MB | 0.31 | 0.46 | 1.09 | 6.47 | 31.21 | 42.45 | 178.07 | 37.15 |
| | 128MB | 0.31 | 0.60 | 1.62 | 9.10 | 34.62 | 56.03 | 229.31 | 47.37 |
| | 256MB | 0.32 | 0.62 | 1.82 | 10.25 | 38.28 | 62.67 | 256.22 | 52.88 |
| | 512MB | 0.33 | 0.68 | 2.05 | 10.59 | 40.55 | 65.74 | 272.23 | 56.03 |
| ROAST | 4MB | 0.28 | 0.30 | 0.27 | 0.48 | 0.99 | 1.36 | 4.83 | 1.22 |
| | 32MB | 0.28 | 0.29 | 0.27 | 0.44 | 1.01 | 1.38 | 4.88 | 1.22 |
| | 64MB | 0.28 | 0.29 | 0.27 | 0.44 | 1.00 | 1.40 | 4.93 | 1.23 |
| | 128MB | 0.30 | 0.27 | 0.27 | 0.45 | 1.01 | 1.39 | 4.91 | 1.23 |
| | 256MB | 0.30 | 0.27 | 0.27 | 0.44 | 1.01 | 1.40 | 4.90 | 1.23 |
| | 512MB | 0.30 | 0.30 | 0.27 | 0.45 | 1.02 | 1.39 | 4.95 | 1.24 |

## C  ROAST-MM LATENCY MEASUREMENTS

### C.1  INFERENCE OPTIMIZATION

### C.2  TRAINING OPTIMIZATION

See tables 8, 9, 10, 11

| | | forward(ms) (optimized for forward + backward) | | | | | | | |
|---|---|---|---|---|---|---|---|---|---|
| | | dim (Matrix dimension = dim x dim) | | | | | | | |
| | Memory (mb) | 512 | 1024 | 2048 | 4096 | 8096 | 10240 | 20480 | Average |
| Full (uncompressed) | | 0.16 | 0.12 | 0.12 | 0.24 | 0.66 | 0.91 | 3.03 | 0.75 |
| HashedNet | 4 | 0.37 | 0.35 | 0.65 | 2.04 | 6.23 | 9.62 | 35.64 | 7.84 |
| | 32 | 0.39 | 0.42 | 0.90 | 3.67 | 13.73 | 22.06 | 92.83 | 19.14 |
| | 64 | 0.33 | 0.47 | 1.11 | 6.45 | 25.78 | 42.51 | 178.20 | 36.41 |
| | 128 | 0.28 | 0.56 | 1.61 | 9.07 | 34.21 | 56.07 | 229.34 | 47.31 |
| | 256 | 0.20 | 0.54 | 1.72 | 9.95 | 38.17 | 62.47 | 258.11 | 53.02 |
| | 512 | 0.14 | 0.50 | 1.88 | 10.37 | 40.40 | 65.43 | 272.19 | 55.84 |
| ROAST | 4 | 0.30 | 0.31 | 0.31 | 0.50 | 1.43 | 2.01 | 7.54 | 1.77 |
| | 32 | 0.30 | 0.33 | 0.35 | 0.55 | 1.44 | 2.09 | 7.59 | 1.81 |
| | 64 | 0.29 | 0.31 | 0.33 | 0.56 | 1.45 | 2.08 | 7.80 | 1.83 |
| | 128 | 0.25 | 0.27 | 0.28 | 0.54 | 1.41 | 2.09 | 7.84 | 1.81 |
| | 256 | 0.16 | 0.18 | 0.19 | 0.46 | 1.33 | 2.02 | 7.82 | 1.74 |
| | 512 | 0.21 | 0.06 | 0.13 | 0.41 | 1.29 | 1.97 | 4.98 | 1.29 |

Table 8: Inference (forward pass time) for different shapes of square weight matrix with input batch of 512. The tile-parameters of multiplication are optimized for each function over "forward + backward" pass .The measurements are taken with tf32 on A100 (48GB)

## D  VARIANCE IN QUALITY OVER DIFFERENT RUNS

The figure 5 shows three runs of ROASTed BERT and BERT models

| | Memory (mb) | backward(ms) (optimized for forward + backward) | | | | | | | |
|---|---|---|---|---|---|---|---|---|---|
| | | dim (Matrix dimension = dim x dim) | | | | | | | |
| | | 512 | 1024 | 2048 | 4096 | 8096 | 10240 | 20480 | Average |
| Full (uncompressed) | | 0.35 | 0.22 | 0.24 | 0.48 | 1.35 | 2.01 | 7.65 | 1.76 |
| HashedNet | 4 | 0.65 | 0.53 | 0.95 | 2.60 | 8.51 | 13.21 | 56.59 | 11.86 |
| | 32 | 0.68 | 0.69 | 1.80 | 6.36 | 24.13 | 38.95 | 160.54 | 33.31 |
| | 64 | 0.74 | 1.06 | 2.81 | 10.78 | 41.35 | 67.02 | 271.86 | 56.52 |
| | 128 | 0.91 | 1.34 | 3.40 | 12.41 | 51.00 | 81.25 | 337.31 | 69.66 |
| | 256 | 1.29 | 1.84 | 4.02 | 14.57 | 58.03 | 91.18 | 376.83 | 78.25 |
| | 512 | 2.08 | 2.62 | 4.90 | 16.24 | 62.45 | 98.46 | 391.46 | 82.60 |
| ROAST | 4 | 0.54 | 0.54 | 0.60 | 1.20 | 2.54 | 3.72 | 13.99 | 3.30 |
| | 32 | 0.57 | 0.61 | 0.69 | 1.06 | 2.71 | 4.04 | 15.07 | 3.54 |
| | 64 | 0.64 | 0.73 | 0.77 | 1.17 | 2.82 | 4.18 | 15.50 | 3.69 |
| | 128 | 0.79 | 0.81 | 0.89 | 1.38 | 3.17 | 4.73 | 18.30 | 4.30 |
| | 256 | 1.19 | 1.17 | 1.27 | 1.77 | 3.56 | 5.17 | 18.33 | 4.64 |
| | 512 | 2.11 | 1.92 | 2.12 | 2.53 | 4.33 | 5.98 | 22.71 | 5.96 |

Table 9: Backward pass for different shapes of square weight matrix with input batch of 512. The tile-parameters of multiplication are optimized for each function over "forward + backward" pass .The measurements are taken with tf32 on A100 (48GB)

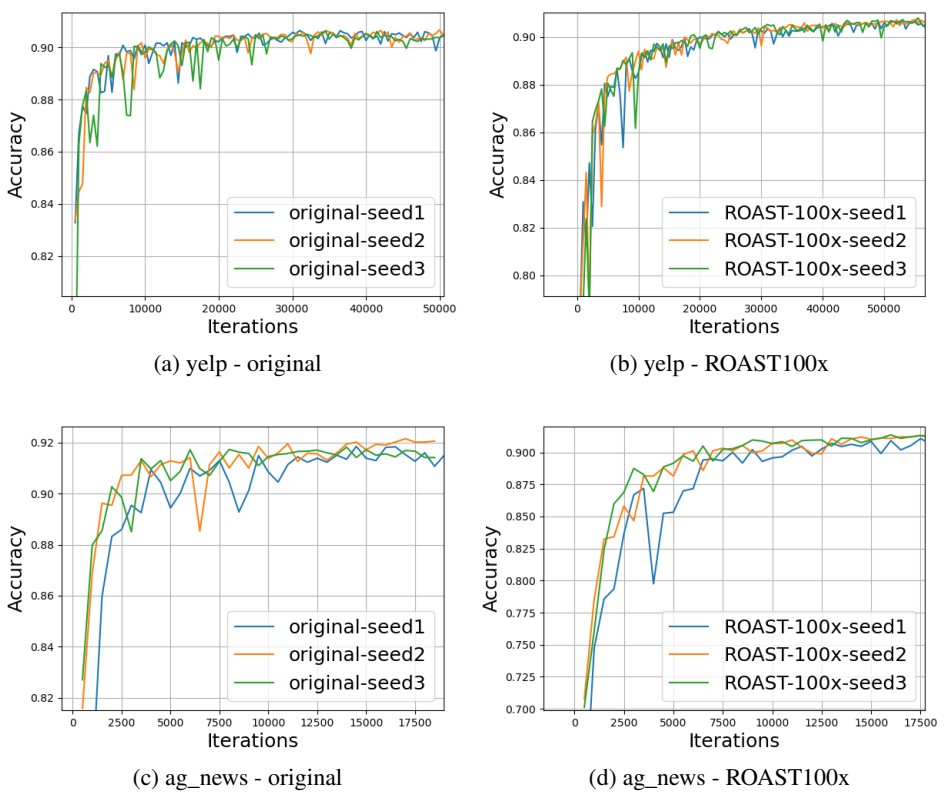

(a) yelp - original

(b) yelp - ROAST100x

(c) ag_news - original

(d) ag_news - ROAST100x

Figure 5: Three runs of original and ROAST-100x runs

| optim | Model | msize | update weights (optim.step())(ms) (optimized for forward + backward) | | | | | | | |
| | | | dim (Matrix dimension = dim x dim) | | | | | | | |
| | | | 512 | 1024 | 2048 | 4096 | 8096 | 10240 | 20480 | Average |
| adagrad | Full | | 0.14 | 0.11 | 0.15 | 0.60 | 2.16 | 3.41 | 13.45 | 2.86 |
| | HashedNet | 4 | 0.14 | 0.11 | 0.11 | 0.11 | 0.11 | 0.12 | 0.54 | 0.18 |
| | | 32 | 0.35 | 0.33 | 0.33 | 0.33 | 0.33 | 0.34 | 0.36 | 0.34 |
| | | 64 | 0.61 | 0.61 | 0.61 | 0.61 | 0.61 | 0.62 | 0.61 | 0.61 |
| | | 128 | 1.15 | 1.14 | 1.14 | 1.14 | 1.15 | 1.19 | 1.18 | 1.15 |
| | | 256 | 2.22 | 2.21 | 2.21 | 2.21 | 2.22 | 2.26 | 3.87 | 2.46 |
| | | 512 | 4.36 | 4.36 | 4.35 | 4.35 | 4.37 | 4.40 | 4.47 | 4.38 |
| | ROAST | 4 | 0.11 | 0.11 | 0.11 | 0.11 | 0.12 | 0.11 | 0.11 | 0.11 |
| | | 32 | 0.33 | 0.34 | 0.34 | 0.33 | 0.33 | 0.33 | 0.33 | 0.33 |
| | | 64 | 0.60 | 0.61 | 0.61 | 0.60 | 0.61 | 0.61 | 0.61 | 0.61 |
| | | 128 | 1.14 | 1.14 | 1.14 | 1.14 | 1.14 | 1.14 | 1.14 | 1.14 |
| | | 256 | 2.21 | 2.21 | 2.21 | 2.21 | 2.21 | 2.21 | 2.21 | 2.21 |
| | | 512 | 4.38 | 4.35 | 4.36 | 4.35 | 4.35 | 4.36 | 4.35 | 4.36 |
| adam | Full | | 0.15 | 0.15 | 0.23 | 1.06 | 3.89 | 6.18 | 24.47 | 5.16 |
| | HashedNet | 4 | 0.15 | 0.23 | 0.16 | 0.16 | 0.16 | 0.16 | 0.16 | 0.17 |
| | | 32 | 0.57 | 0.57 | 0.57 | 0.57 | 0.57 | 0.57 | 0.59 | 0.57 |
| | | 64 | 1.06 | 1.06 | 1.06 | 1.06 | 1.06 | 1.06 | 1.16 | 1.08 |
| | | 128 | 2.03 | 2.05 | 2.04 | 2.04 | 2.05 | 2.04 | 2.23 | 2.07 |
| | | 256 | 3.98 | 3.99 | 3.98 | 3.99 | 4.00 | 4.00 | 4.22 | 4.02 |
| | | 512 | 7.89 | 7.89 | 7.89 | 7.89 | 7.91 | 7.90 | 8.13 | 7.93 |
| | ROAST | 4 | 0.15 | 0.23 | 0.15 | 0.16 | 0.16 | 0.15 | 0.16 | 0.17 |
| | | 32 | 0.57 | 0.57 | 0.57 | 0.57 | 0.57 | 0.57 | 0.57 | 0.57 |
| | | 64 | 1.07 | 1.06 | 1.06 | 1.06 | 1.06 | 1.07 | 1.06 | 1.06 |
| | | 128 | 2.05 | 2.03 | 2.04 | 2.04 | 2.03 | 2.04 | 2.04 | 2.04 |
| | | 256 | 4.01 | 3.98 | 3.99 | 3.99 | 3.99 | 3.99 | 3.99 | 3.99 |
| | | 512 | 7.89 | 7.89 | 7.89 | 7.89 | 7.89 | 7.89 | 7.89 | 7.89 |
| sgd | Full | | 0.08 | 0.07 | 0.08 | 0.20 | 0.62 | 0.97 | 3.92 | 0.85 |
| | HashedNet | 4 | 0.08 | 0.07 | 0.08 | 0.07 | 0.07 | 0.08 | 0.08 | 0.08 |
| | | 32 | 0.12 | 0.12 | 0.12 | 0.12 | 0.12 | 0.12 | 0.17 | 0.13 |
| | | 64 | 0.19 | 0.20 | 0.20 | 0.20 | 0.20 | 0.21 | 0.31 | 0.22 |
| | | 128 | 0.35 | 0.34 | 0.34 | 0.35 | 0.35 | 0.37 | 0.48 | 0.37 |
| | | 256 | 0.64 | 0.64 | 0.64 | 0.64 | 0.65 | 0.67 | 0.83 | 0.67 |
| | | 512 | 1.23 | 1.23 | 1.23 | 1.23 | 1.25 | 1.24 | 1.25 | 1.24 |
| | ROAST | 4 | 0.07 | 0.07 | 0.07 | 0.08 | 0.07 | 0.07 | 0.23 | 0.10 |
| | | 32 | 0.12 | 0.12 | 0.13 | 0.12 | 0.12 | 0.12 | 0.12 | 0.12 |
| | | 64 | 0.22 | 0.19 | 0.20 | 0.19 | 0.19 | 0.20 | 0.29 | 0.21 |
| | | 128 | 0.34 | 0.35 | 0.34 | 0.34 | 0.34 | 0.35 | 0.40 | 0.35 |
| | | 256 | 0.64 | 0.65 | 0.64 | 0.64 | 0.64 | 0.65 | 0.64 | 0.64 |
| | | 512 | 1.27 | 1.23 | 1.23 | 1.23 | 1.28 | 1.23 | 1.62 | 1.30 |

Table 10: Weight update operation (optimizer.step()) for different shapes of square weight matrix with input batch of 512. The tile-parameters of multiplication are optimized for each function over "forward + backward" pass .The measurements are taken with tf32 on A100 (48GB)

| optim | Model | msize | total = fwd + bkwd + optimize (ms) (optimized for forward + backward) | | | | | | | |
|---|---|---|---|---|---|---|---|---|---|---|
| | | | dim (Matrix dimension = dim x dim) | | | | | | | |
| | | | 512 | 1024 | 2048 | 4096 | 8096 | 10240 | 20480 | Average |
| adagrad | Full | | 0.65 | 0.46 | 0.51 | 1.32 | 4.17 | 6.33 | 24.13 | 5.37 |
| | HashedNet | 4 | 1.16 | 0.99 | 1.71 | 4.74 | 14.86 | 22.95 | 92.78 | 19.88 |
| | | 32 | 1.43 | 1.44 | 3.03 | 10.37 | 38.19 | 61.35 | 253.72 | 52.79 |
| | | 64 | 1.68 | 2.14 | 4.53 | 17.83 | 67.74 | 110.15 | 450.66 | 93.53 |
| | | 128 | 2.34 | 3.04 | 6.15 | 22.62 | 86.36 | 138.51 | 567.83 | 118.12 |
| | | 256 | 3.71 | 4.59 | 7.95 | 26.73 | 98.42 | 155.92 | 638.80 | 133.73 |
| | | 512 | 6.58 | 7.47 | 11.13 | 30.96 | 107.21 | 168.30 | 668.12 | 142.83 |
| | ROAST | 4 | 0.95 | 0.95 | 1.02 | 1.81 | 4.09 | 5.84 | 21.64 | 5.19 |
| | | 32 | 1.21 | 1.27 | 1.38 | 1.94 | 4.49 | 6.46 | 23.00 | 5.68 |
| | | 64 | 1.54 | 1.64 | 1.70 | 2.34 | 4.87 | 6.86 | 23.90 | 6.12 |
| | | 128 | 2.18 | 2.22 | 2.31 | 3.06 | 5.72 | 7.97 | 27.28 | 7.25 |
| | | 256 | 3.57 | 3.56 | 3.67 | 4.43 | 7.10 | 9.40 | 28.35 | 8.58 |
| | | 512 | 6.70 | 6.32 | 6.62 | 7.29 | 9.97 | 12.31 | 32.04 | 11.61 |
| adam | Full | | 0.50 | 0.48 | 0.60 | 1.78 | 5.89 | 9.11 | 35.01 | 7.62 |
| | HashedNet | 4 | 1.00 | 1.56 | 1.76 | 4.81 | 14.94 | 23.07 | 86.76 | 19.13 |
| | | 32 | 1.43 | 1.78 | 3.29 | 10.60 | 38.45 | 61.64 | 253.20 | 52.91 |
| | | 64 | 2.03 | 2.63 | 4.97 | 18.35 | 68.28 | 110.63 | 450.86 | 93.96 |
| | | 128 | 3.18 | 4.27 | 7.02 | 23.54 | 87.47 | 139.30 | 568.72 | 119.07 |
| | | 256 | 5.45 | 6.30 | 9.71 | 28.66 | 100.19 | 157.55 | 633.80 | 134.52 |
| | | 512 | 10.08 | 10.94 | 14.64 | 34.56 | 110.71 | 171.67 | 672.24 | 146.41 |
| | ROAST | 4 | 1.00 | 1.27 | 1.05 | 1.86 | 4.06 | 5.89 | 21.71 | 5.26 |
| | | 32 | 1.45 | 1.56 | 1.52 | 2.21 | 4.72 | 6.69 | 23.28 | 5.92 |
| | | 64 | 2.13 | 2.02 | 2.18 | 2.80 | 5.34 | 7.39 | 24.35 | 6.60 |
| | | 128 | 3.26 | 3.11 | 3.23 | 3.95 | 6.62 | 8.85 | 28.22 | 8.18 |
| | | 256 | 5.82 | 5.33 | 5.45 | 6.21 | 8.97 | 11.15 | 30.19 | 10.45 |
| | | 512 | 9.82 | 9.87 | 10.14 | 10.90 | 13.52 | 15.82 | 35.59 | 15.09 |
| sgd | Full | | 0.44 | 0.43 | 0.46 | 0.90 | 2.62 | 3.90 | 14.68 | 3.35 |
| | HashedNet | 4 | 1.25 | 0.95 | 1.70 | 4.72 | 14.76 | 22.96 | 86.70 | 19.01 |
| | | 32 | 0.99 | 1.23 | 2.86 | 10.17 | 38.10 | 61.16 | 252.99 | 52.50 |
| | | 64 | 1.16 | 1.84 | 4.11 | 17.51 | 67.28 | 109.78 | 450.34 | 93.15 |
| | | 128 | 1.59 | 2.24 | 5.28 | 21.84 | 85.46 | 137.54 | 566.88 | 117.26 |
| | | 256 | 2.21 | 3.00 | 6.35 | 25.19 | 96.91 | 154.43 | 630.75 | 131.26 |
| | | 512 | 3.42 | 4.28 | 8.06 | 27.91 | 104.03 | 164.94 | 665.29 | 139.70 |
| | ROAST | 4 | 0.92 | 0.92 | 0.98 | 1.79 | 3.94 | 5.82 | 22.39 | 5.25 |
| | | 32 | 0.95 | 1.00 | 1.17 | 1.75 | 4.28 | 6.25 | 22.77 | 5.45 |
| | | 64 | 1.62 | 1.15 | 1.26 | 1.92 | 4.45 | 6.45 | 24.01 | 5.84 |
| | | 128 | 1.38 | 1.44 | 1.52 | 2.26 | 4.90 | 7.25 | 27.18 | 6.56 |
| | | 256 | 2.04 | 2.10 | 2.14 | 2.85 | 5.53 | 7.91 | 26.98 | 7.08 |
| | | 512 | 3.56 | 3.20 | 3.36 | 4.18 | 7.10 | 9.17 | 31.20 | 8.82 |

Table 11: Total training step time for different shapes of square weight matrix with input batch of 512. The tile-parameters of multiplication are optimized for each function over "forward + backward" pass .The measurements are taken with tf32 on A100 (48GB)

