# OpenReview forum: "Hardware-aware compression with Random Operation Access Specific Tile (ROAST) hashing"
_ICLR.cc/2023/Conference — Submitted to ICLR 2023_

### Official Review · Reviewer_sRtG · 2022-10-23

**Confidence:** 3
**Clarity, Quality, Novelty And Reproducibility:** Please see the questions above.
**Correctness:** 3
**Technical Novelty And Significance:** 2
**Empirical Novelty And Significance:** 2
**Recommendation:** 5

**Strength And Weaknesses:**

## Strength
1. The proposed ROAST achieves similar quality in almost 100x less space for compression model training. ROAST performs well in many research areas including text classification and image classification. The experimental results show that the proposed global memory sharing method performs better than previous local memory-sharing.

## Weakness
1. Lack of comparison of time costs between GMS and LMS. Will this global memory share cause access conflict?

2. For Local vs. Global memory sharing in Figure 3(b), the GMS shows more significant accuracy degradation than LMS. How do you know this is not relevant to the GMS method but to other factors?

3. Lack of experiments on different models and datasets. Many text-classification tasks are introduced in the introduction but only experiment on BERT-2-2 and BERT-base. For the image classification task, only ResNet-9 on CIFAR-10 is used.

4. Need more details about the implementation of ROAST operations. It is hard to follow.


**Summary Of The Paper:**

This paper proposes a model compression approach: Random Operation Access Specific Tile (ROAST) hashing. The authors consider three operations for block-based hashing to reduce memory usage, and they also introduce a global memory-sharing method to improve model accuracy. Experiments on BERT and ResNet show that the proposed ROAST exceeds HashedNet and achieves high compression when training models.

**Summary Of The Review:**

While the paper proposes an interesting idea, the experiments are insufficient, and some details of the proposed technique are missing.

---

> ### Author Response · Authors · 2022-11-09
> **Response to questions/concerns**
>
> "Lack of comparison of time costs between GMS and LMS. Will this global memory share cause access conflict?"
>
> Implementation wise GMS and LMS are essentially the same computation. Hence, the forward pass and backward pass look exactly the same. The only difference is that  GMS accesses the entire ROAST array as opposed to the module specific array in LMS. In terms of “access conflict” or alternatively collisions, as we show in section 5 (theory), GMS actually enables lower collisions as even though each memory location is accessible to many more parameters, every parameter can also choose to access a larger range of memory. The overall effect is lower collisions and better dimensionality reduction (section 5 theorem).
>
> In terms of convergence, we add results that reveal that GMS actually converges faster than LMS. See newly added table 6 in appendix. For example, in amazon-polarity in order to achieve the target accuracy of 93.4, 10x-GMS needs only 1.97 epochs whereas 10x-LMS needs 3.77 epochs (see table 6 in appendix attached to main paper). Similar results can be seen for other compression and dataset settings.
>
> "For Local vs. Global memory sharing in Figure 3(b), the GMS shows more significant accuracy degradation than LMS. How do you know this is not relevant to the GMS method but to other factors?"
>
> We found a small issue (inconsequential to overall conclusion , changes the conclusion in favor of ROAST-GMS ) We found that there was an issue with this plot. The main issue was that GMS was run with a different batch size 64 (due to a typo in our config) and all others were run with batch size of 32. However, the plot was generated w.r.t. Iterations so it was a weird comparison to look at. We have fixed this in the original figure and added separate figures in appendix A.2 (see figure 4) . The plot was weird because due to the batch size and plotting issue, we were looking at 2X epochs for GMS and only X epochs of LMS stretched to the same scale. The updated plots with 10 epochs for all the models give a clearer picture. ( see figure 4 in the appendix attached to the main paper)
>
> "Lack of experiments on different models and datasets. Many text-classification tasks are introduced in the introduction but only experiment on BERT-2-2 and BERT-base. For the image classification task, only ResNet-9 on CIFAR-10 is used."
>
> In the paper, we use two of the largest text-classification datasets with BERT architecture and CIFAR 10 on the convolution architecture. The baselines are also compared on the same datasets. We show that ROAST is a better technique to efficiently train models with lower memory than existing approaches of pruning/quantization or hashednets. Also, they give better / competitive compression rates on the NLP and vision tasks compared to pruning or architecture reduction approaches. Also, they have the potential to give much more compression than what other methods, like quantization etc, can give. (for example, we can compress the BERT model $100\times$ when it suits the task, but methods like quantization have an upper limit of $32\times$). Additionally, ROAST enables memory reduction at the training stage itself, not requiring the space to store the entire original model. Furthermore, ROAST can be combined with these orthogonal approaches of pruning and quantization to obtain best of both worlds.
>
> "Need more details about the implementation of ROAST operations. It is hard to follow."
>
> We explain the ROAST operations details in 4.1 and provide the kernel implementation code for the same. Any advice on how we can make the explanation better is welcome. We will try to incorporate it in our next version.

---

### Official Review · Reviewer_rkTz · 2022-10-29

**Confidence:** 3
**Correctness:** 3
**Technical Novelty And Significance:** 3
**Empirical Novelty And Significance:** 2
**Recommendation:** 5

**Clarity, Quality, Novelty And Reproducibility:**

Questions related to quality of results are discussed in strengths and weakness section.

**Strength And Weaknesses:**

Section 2 and 3 provide a good overview of the targeted problem space and the immediate baselines (ROBE/HashedNet) which are closest to the proposed ideas. The diagrams provided in the paper are helpful in explaining the proposed ideas and authors also present detailed proofs that help justify the superiority of GMS approach over LMS. Authors show large speed up compared to the previous hashing approach, HashedNet.

In Section 6, authors explain the experimental setup designed to verify the efficacy of ROAST. While the authors explain that they perform some hyperparameter tuning to achieve a given target accuracy it fails to communicate how using GMS approach in ROAST impacts the trainability or ease of training of their models. E.g., Adding the epochs of training incurred to achieve the target accuracy for the different baselines in Table 3 and 4, might help explain that. The intuition is this, since the weights are shared across all layers the cumulative gradient update application might make it harder to get the same level of accuracy with fewer unique weights (compared to LMS).

In Table 4, authors demonstrate that ROAST can be combined with other model compression techniques e.g., PRUNING to achieve further model compression. While this is lucrative, structured pruning that takes advantage of the block-wise nature of ROAST memory access could achieve even better accuracy with low training effort. Consider trying that as well.

Table 5 shows the inference runtime for a range of memory size and model size values on a GPU with 48MB cache. While the ROAST inference time dominates over HashedNet baseline, especially for large model sizes (high compression ratio). The comparison with PyTorch based matrix multiplication (MM) is also interesting. While PyTorch with its efficient matrix multiply implementation achieves fast inference time for lower model sizes, as the model size grows the benefit of PyTorch over ROAST reduces. Hinting potentially that ROAST based memory optimization could further help improve PyTorch MM implementation for hashed DNNs.
The results in Figure 3b are somewhat contradictory to that result that GMS is superior to LMS for a range of models. Considering that the Local ROAST approaches (10x and 100x) end on a slightly higher accuracy number than corresponding GMS approaches. Even seeing accuracy values that match closely for GMS and LMS approaches is surprising. Kindly explain why the yelp-polarity model defies the expected trend as seen in Figure 3a.

I appreciate the authors clearly stating the limitations of the proposed work (focus on reducing memory access during training/inference only, not compute cost)


**Summary Of The Paper:**


The paper presents ROAST, a hashing-based technique for reducing the memory usage of ML models for both training and inference. ROAST improves on the previously existing hashing techniques for this purpose (ROBE/HashedNet) by using block-based (cache/HW – friendly) memory accesses. It also proposes the policy of globally sharing weights (across layers) as opposed to prior art (HashedNet) which shared weights per layer only. This in turn allows ROAST to achieve much higher compression ratios, which translate to much larger performance gains with much lower memory footprint over the state-of-the-art techniques.

The authors provide theoretical and empirical proof showing that their global memory sharing approach is superior to local memory sharing used in prior art. Lastly, authors demonstrate strong accuracy results using ROAST for a range of Text classification and Image classification benchmarks.


**Summary Of The Review:**

The results show in Figure 3b) violate the statement proved showing GMS should achieve higher performance than LMS. Without a good justification for this miss, it is hard to confirm that the GMS technique has been effectively scaled to all DNN layers. Further, it is important to understand if the GMS approach brings with it any addition to training complexity (e.g., increased hyperparameter tuning). If the authors can add data for these requests, it will help consolidate the presented results.

---

> ### Author Response · Authors · 2022-11-09
> **response to concerns / questions.**
>
> "In Section 6, authors explain the experimental setup designed to verify the efficacy....(GMS vs LMS how does it affect the learning)"
>
> That is a great suggestion. We added an extended version of table 3 ( see table 6 in appendix). The following are our findings.  As we can see in table 6, across the board (10x-100x compressions and yelp and amazon datasets), GMS actually converges to the target accuracy faster than LMS. This showcases that GMS is at an advantage when it comes to learning than LMS.
>
> Along the same lines as what the reviewer is hinting at, we can look at the number of ‘unique weights’ or equivalently a softer criteria : number of collisions for each parameter in ROAST array. This is exactly what is captured in our theory in section 5. And even theoretically, the load of collisions on each parameter reduces in GMS. So even in this sense we expect GMS to be better for learning than LMS. So our empirical results align with the theoretical expectations.
>
> "In Table 4, authors demonstrate that ROAST...(structured pruning with roast) " "Table 5 shows the inference runtime .. (Scaling efficiency of ROAST for general pytorch-MM)"
>
> These are great suggestions for future work.
>
> "The results in Figure 3b are somewhat contradictory... (the weird plot 3b) "
>
> We found a small issue (inconsequential to overall conclusion, changes the conclusion in favor of ROAST-GMS ) We found that there was an issue with this plot. The main issue was that GMS was run with a different batch size 64 (due to a typo in our config) and all others were run with batch size of 32. However, the plot was generated w.r.t. Iterations so it was a weird comparison to look at. We have fixed this in the original figure and added separate figures in appendix A.2 (see figure 4) . The plot was weird because due to the batch size and plotting issue, we were looking at 2X epochs for GMS and only X epochs of LMS stretched to the same scale. The updated plots with 10 epochs for all the models give a clearer picture. (See figure 4 in the appendix attached to the main paper)

---

### Official Review · Reviewer_4EgW · 2022-10-30

**Confidence:** 4
**Clarity, Quality, Novelty And Reproducibility:** The quality of the paper is good. The…
**Correctness:** 3
**Technical Novelty And Significance:** 3
**Empirical Novelty And Significance:** 3
**Recommendation:** 5

**Strength And Weaknesses:**

Strength: Overall, this paper is well organised. Experiment results has well proved the efficiency of the method

Weakness: 1.	Weight sharing can lower the memory usage but can not reduce the number of operations.  As authors claimed, the RAM access is around 100X slower than the computation. This is true. However, from the hardware perspective, a efficient way to lower the memory access delay is cache. The real memory access delay can be much shorter, and can be almost the same as the computation time. Besides, as the computation of neural network is regular, we can use pipelined design to further eliminate the negative impact of memory access. Besides, with the cache design, we can substantially reduce the energy consumption of memory access. I understand this is the common problem of memory sharing, not the problem of this paper only. However, it would be better to do a more comprehensive comparison with other model compression methods not only on model size but also on latency and energy consumption.

2.	The comparison is not enough. In terms of weight sharing, please do a more comprehensive comparison with other state-of-the-art weight sharing methods.

3.	Does table 5 compared under the same accuracy?

**Summary Of The Paper:**

Weight sharing is one of the effective way to compress the model. This paper propose a new hash method to lower the memory usage for neural network models. Compare with the previous hash method, which use the local memory sharing, this paper propose a new idea call global memory sharing, which is aimed to lower the memory usage.





**Summary Of The Review:**

Overall, the paper's idea is good in the area of weight sharing. The problem of this paper is the lack of a comprehensive comparison with state-of-the-art. At least it should be compared with more weight sharing methods.

When compared with model compression method like pruning, the author only list the model size. It is better to also compare the energy consumption and latency.

---

> ### Author Response · Authors · 2022-11-09
> **Response to concerns / comments.**
>
> "Response to weakness 1. The reviewer makes two points (1) There might be ways to handle memory latency in neural networks which also reduces computational costs (2) energy consumption, computational workload and latency should also be compared. "
>
> We believe that engineering solutions like (1) might be applicable to even ROAST based models and might perform better with lower memory footprints. As for (2), as we state in our last section of introduction, this paper focuses on the memory reduction approach and thus does not evaluate energy and computations. We completely agree that these are important considerations and are of utmost importance for us. They are a great future direction for this work.  In order to showcase one of our key contributions - improving efficiency of parameter sharing methods, we implement HashedNet and ROAST in the same system and show that ROAST is much faster than HashedNet in terms of latency.
>
> "The comparison is not enough. In terms of weight sharing, please do a more comprehensive comparison with other state-of-the-art weight sharing methods."
>
> As it turns out, model-agnostic parameter-sharing methods have yet to be widely explored in the literature. It is evident in the following recent surveys [1,2,3,4,5]. 3/5 surveys do not talk about parameter-sharing methods, likely because parameter-sharing is not widely used in popular application domains of NLP, vision, etc. 2/5 surveys talk about parameter-sharing. The only model-agnostic parameter sharing they talk about is HashedNets. Other parameter-sharing methods are model/ module specific. For example, SlimEmbeddings, character-aware-embeddings for embedding tables, layer sharing for transformer models, encoder-decoder sharing for translation models, and so on. The methods, such as layer sharing etc., can be potentially applied to our experiments. However, they will only give a small factor of compression. For example, in BERT-2-2, layer sharing will only give us 2$\times$ compression. One of the reasons why parameter-sharing methods like HashedNets are not widely explored is the associated efficiency issue. With ROAST, we are resolving the efficiency issues related to HashedNets and improving it in a theoretically sound manner. We strongly believe that parameter-sharing methods have immense value, and with ROAST, we are getting closer to realizing it.
>
> [1]Cheng, Y., Wang, D., Zhou, P. and Zhang, T., 2017. A survey of model compression and acceleration for deep neural networks. arXiv preprint arXiv:1710.09282.
>
> [2]Gupta, M. and Agrawal, P., 2022. Compression of deep learning models for text: A survey. ACM Transactions on Knowledge Discovery from Data (TKDD), 16(4), pp.1-55
>
> [3] Menghani, G., 2021. Efficient deep learning: A survey on making deep learning models smaller, faster, and better. arXiv preprint arXiv:2106.08962
>
> [4] Xu, C. and McAuley, J., 2022. A survey on model compression for natural language processing. arXiv preprint arXiv:2202.07105.
>
> [5] chester256/Model-Compression-Papers: Papers for deep neural network compression and acceleration (github.com)
>
>
> "Does table 5 compared under the same accuracy?"
>
> Table 5 compares the latency of algorithmic implementation of matrix multiplication using full matrix multiplication, using HashedNet-style weight sharing and ROAST style hardware aware weight sharing. It is not associated with a learning process.

---

> > ### Comment · Reviewer_4EgW · 2022-11-25
> > **comment**
> >
> > The authors added more state-of-the-art work, which improved the quality of this paper. However, I think the scope of this paper is still limited because the metrics like energy consumption were not included. Hence, I keep the original score.

---

### Official Review · Reviewer_MkTt · 2022-11-02

**Confidence:** 4
**Correctness:** 2
**Technical Novelty And Significance:** 2
**Empirical Novelty And Significance:** 3
**Recommendation:** 5

**Clarity, Quality, Novelty And Reproducibility:**

The paper is clearly written for the most part. This work builds on the idea of ROBE, but expands its scope to a generalized embedding lookup operation and non-embedding operations.

**Strength And Weaknesses:**

**Strengths**

* 100$\times$ reduction of memory usage with no accuracy drop is impressive.

* ROAST addresses the memory usage in both training and serving.

* It is interesting to see only three ROAST operations are sufficient for running DL models.

**Weaknesses**

* ROAST has been evaluated on relatively simple tasks with a small number of classes and it is not clear how it is applicable to larger, more complex tasks without causing accuracy drop.

* The baseline used for evaluation is HashedNet, which seems somewhat outdated.

* In ROAST-MM there is a model-specific parameter $\lambda$, which is determined by "some constant $C$", and it is not clear how sensitive the performance is to the setting of this parameter.


**Summary Of The Paper:**

This paper introduces ROAST (Random Operation Access Specific Tile) hashing, a model-agnostic, hardware-aware model compression framework. ROAST essentially provides a global parameter sharing method to give arbitrary control to the user over the memory footprint of model during both training and inference. Evaluation with both BERT and ResNet-9 demonstrates the feasibility of 100x memory footprint reduction without accuracy degradation.


**Summary Of The Review:**

As the NN scales, there are growing concerns for efficient resource usage for both training and inference--in particular for memory capacity and bandwidth. That said, this paper tackles a timely problem to mitigate this memory bottelenck. On the positive side, the proposed hardware-friendly paramter sharing scheme looks promising and presents impressive results for both BERT-2-2 and ResNet-9. Also, the authors address the memory capacity problem not only for inference (which most existing works tackle) but also for training.

However, I still have several concerns about this work, at least in its current form, as follows:

* ROAST is evaluated on relatively simple classification tasks with a small number of classes. It would have made this work much stronger if the authors evaluate more complex tasks such as ImageNet, SQuAD, and so on.  At the end of the day, the proposed framework would be most useful for large models that can handle complex tasks, but evaluation is currently falling short. I am wondering how ROAST would perform on those complex tasks.

* The baseline used for evaluation is HashedNet, which was introduced in 2015 and hence seems outdated. Is there any stronger, more SOTA baselines to compare against?  How would ROAST compare against them?

* ROAST-MM introduces a scaling factor $\lambda$, which is dependent upon "some constant $C$". It is not clear to me how to set this parameter. Also, how sensitive is the performance of ROAST to the setting of this parameter? How much effort is needed to set this parameter properly? What factor does affect the setting of this parameter? Model? Hardware configuration (like cache size)? Or, both?  Please elaborate.

---

> ### Author Response · Authors · 2022-11-09
> **response to concerns /  questions**
>
>
> "ROAST has been evaluated on relatively simple tasks with a small number of classes and it is not clear how it is applicable to larger, more complex tasks without causing accuracy drop."
>
> In the paper, we use two of the largest text-classification datasets with BERT architecture and CIFAR 10 on the convolution architecture. The baselines are also compared on the same datasets. We show that ROAST is a better technique to efficiently train models with lower memory than existing approaches of pruning/quantization or hashednets. Also, they give better / competitive compression rates on the NLP and vision tasks compared to pruning or architecture reduction approaches. Also, they have the potential to give much more compression than what other methods, like quantization etc, can give. (for example, we can compress the BERT model 100$\times$ when it suits the task, but methods like quantization have an upper limit of 32$\times$). Additionally, ROAST enables memory reduction at the training stage itself, not requiring the space to store the entire original model. Furthermore, ROAST can be combined with the orthogonal approaches of pruning and quantization to obtain the best of both worlds.
>
> "The baseline used for evaluation is HashedNet, which seems somewhat outdated."
>
> As can be seen from the recent surveys [1,2,3,4,5], the popular approaches for compression are pruning, quantization, and parameter-sharing.
>
> We compare ROAST with pruning on both NLP and vision tasks. We added pruning results to the NLP task in the appendix (table 6), where we observe that ROAST outperforms pruning w.r.t quality of the model achieved for large compressions. Note that ROAST and pruning can be combined to get an even lesser memory footprint.
>
> Quantization can only give a maximum compression of 32$\times$ (usually smaller than this). Thus, Quantization cannot beat ROAST on these tasks. Again, we can combine Quantization with ROAST for more compression.
>
> In parameter-sharing, the only model-agnostic approach is HashedNets. Other parameter-sharing methods are model/ module specific[2]. For example, SlimEmbeddings, character-aware-embeddings cater to embedding tables. Layer sharing can be used for transformer models, encoder-decoder sharing for translation models, and so on. The methods, such as layer sharing, etc., can be potentially applied to our experiments. However, they will only give a small factor of compression. For example, in BERT-2-2, layer sharing will only give us 2$\times$ compression. One of the reasons why parameter-sharing methods like HashedNets are not widely explored is the associated efficiency issue. With ROAST, we are resolving the efficiency issues related to HashedNets and improving it in a theoretically sound manner. We strongly believe that parameter-sharing methods have immense value, and with ROAST, we are getting closer to realizing it.
>
> [1]Cheng, Y., Wang, D., Zhou, P. and Zhang, T., 2017. A survey of model compression and acceleration for deep neural networks. arXiv preprint arXiv:1710.09282.
> [2]Gupta, M. and Agrawal, P., 2022. Compression of deep learning models for text: A survey. ACM Transactions on Knowledge Discovery from Data (TKDD), 16(4), pp.1-55
> [3] Menghani, G., 2021. Efficient deep learning: A survey on making deep learning models smaller, faster, and better. arXiv preprint arXiv:2106.08962
> [4] Xu, C. and McAuley, J., 2022. A survey on model compression for natural language processing. arXiv preprint arXiv:2202.07105.
> [5] chester256/Model-Compression-Papers: Papers for deep neural network compression and acceleration (github.com)
>
>
> "In ROAST-MM there is a model-specific parameter λ, which is determined by "some constant C", and it is not clear how sensitive the performance is to the setting of this parameter."
>
> In deep learning, the different modules in the model are usually initialized so that all modules maintain the norm of the inputs under transformation. When performing global weight sharing, we need to take special care of this. As mentioned in section 4.2, using a scaler on the top of a single initialized roast array is our way of maintaining this.
>
> How to choose C? You can, in theory, choose any arbitrary C. In practice, we do not perform any hyperparameter tuning on C. We select C to be 100 and use that across our experiments. The roast array is initialized with $U(-1/C, 1/C)$
>
> How is $\lambda$ set? Note that $\lambda$ is not a parameter. It is set using the value of C and the module's shape. For example, in the case of $a \times b$ matrix, we want the initialization to be $(U(- 1 / \sqrt(a), 1/ \sqrt(a))$ and hence to achieve this, we use a $\lambda = C / \sqrt(a)$. Thus, the distribution of each element of the matrix retrieved from shared memory initialized with C is $U(-1/\sqrt(a), 1/\sqrt(a))$. Note that scaler $\lambda$ removes the dependence of the distribution on C.

---

### Author Response · Authors · 2022-11-09
**Changes in the revision 1**

List of changes in first revision
(Updated appendix is attached to the main paper itself)

1.  We fixed a small issue with yelp training. It does not change the conclusion of the paper. In fact, it makes a stronger argument in favor of GMS-ROAST. The GMS training was run using incorrect batch size (typo in our config) (64 instead of 32 which was used for original model and LMS). This made plots weird as we were looking at 2X epochs for GMS vs 1X epochs for LMS and original stretched to the same scale. When fixed it gives better accuracy results (updated table 3 and the plots were updated accordingly) and clearer plots. Also, we added separate plots for different compression rates in appendix A.2 for better clarity.

2. We add pruning baselines for NLP task. (See table 6 in appendix A.1) We will move some results from this table in appendix to the table in main paper in the final version

3. We added LMS and GMS results to table 6. Also, for each result in table 3 (and additional baselines and results) we add the number of epochs required to reach the best accuracy. Additionally, we also note epochs required for target accuracies in the lower part of the table 6 which helps in convergence comparison.

---

### Author Response · Authors · 2022-11-11
**A note to reviewers.**

We thank the reviewers for their time and comments which helps us make our paper better. We hope to have sufficiently addressed reviewers' concerns / questions. If the reviewers are satisfied with our answers, we would urge them to adjust the scores accordingly. In case reviewers have additional concerns, we would be happy to answer them. We would request reviewers to raise additional concerns soon to give us enough time to respond to them meaningfully.

---

### Comment · Area_Chair_rbef · 2022-11-20
**Please update your reviews**

Please make sure that your reviews acknowledge authors’ responses and reflect your current evaluation of the paper. This is particularly important if you didn’t directly engage with the authors during the discussion phase (so the authors don’t know if their response changed your evaluation) or if you expressed an intention to update your rating but did not do so.

Cheers,
AC

---

> ### Comment · Area_Chair_rbef · 2022-11-25
> **Last reminder to update your post-rebuttal comments**
>
> Hi reviewers,
>
> Please read through authors' responses and raise any issue if any by TODAY.
> If no further comments, please adjust your score ASAP.
> I need to make decision very soon.
>
> Thanks,
> AC

---

### Decision · Program_Chairs · 2023-01-20

**Decision:**

Reject

**Justification For Why Not Higher Score:**

More experiments (latency and energy consumption, larger CV datasets) are needed to support the claims.

**Justification For Why Not Lower Score:**

NA

**Metareview: Summary, Strengths And Weaknesses:**

Summary: This paper proposes a model compression approach: Random Operation Access Specific Tile (ROAST) hashing. The authors consider three operations for block-based hashing to reduce memory usage, and they also introduce a global memory-sharing method to improve model accuracy. Experiments on BERT and ResNet show that the proposed ROAST exceeds HashedNet and achieves high compression when training models.

Strength: Overall, this paper is well organised. Experiment results are promising, e.g., (100 reduction of memory usage).

Weakness: The scope of this paper is still limited because the metrics like energy consumption were not included. It would be better to do a more comprehensive comparison with other model compression methods not only on model size but also on latency and energy consumption. The benchmark CV datasets is small scale (i.e., CIFAR10) why not try large-scale CV datasets? No justifications are provided.

All the reviews have the consensus on the down side for this paper.